# GCGNet: Graph-Consistent Generative Network for Time Series Forecasting with Exogenous Variables

**Zhengyu Li,** * **Xiangfei Qiu,** * **Yuhan Zhu, Xingjian Wu, Jilin Hu, Chenjuan Guo, Bin Yang**[†]
East China Normal University
{lizhengyu,xfqiu,yhzhu,xjwu}@stu.ecnu.edu.cn,
{jlhu,cjguo,byang}@dase.ecnu.edu.cn

## Abstract

Exogenous variables offer valuable supplementary information for predicting future endogenous variables. Forecasting with exogenous variables needs to consider both past-to-future dependencies (i.e., temporal correlations) and the influence of exogenous variables on endogenous variables (i.e., channel correlations). This is pivotal when future exogenous variables are available, because they may directly affect the future endogenous variables. Many methods have been proposed for time series forecasting with exogenous variables, focusing on modeling temporal and channel correlations. However, most of them use a two-step strategy, modeling temporal and channel correlations separately, which limits their ability to capture joint correlations across time and channels. Furthermore, in real-world scenarios, time series are frequently affected by various forms of noises, underscoring the critical importance of robustness in such correlations modeling. To address these limitations, we propose **GCGNet**, a **G**raph-**C**onsistent **Ge**nerative **Net**work for time series forecasting with exogenous variables. Specifically, GCGNet first employs a Variational Generator to produce coarse predictions. A Graph Structure Aligner then further guides it by evaluating the consistency between the generated and true correlations, where the correlations are represented as graphs, and are robust to noises. Finally, a Graph Refiner is proposed to refine the predictions to prevent degeneration and improve accuracy. Extensive experiments on 12 real-world datasets demonstrate that GCGNet outperforms state-of-the-art baselines. We make our code and datasets available at https://github.com/decisionintelligence/GCGNet.

## 1 Introduction

Time series forecasting plays a crucial role in many domains, including economics (Hu et al., 2026; 2025a), traffic (Xu et al., 2023; Guo et al., 2014; Ma et al., 2014), health (Wei et al., 2022), energy (Alvarez et al., 2011; Guo et al., 2015), environmental applications (Tian et al., 2025; 2026), industrial manufacturing (Wang et al., 2023b; 2026a; Cheng et al., 2026), and AIOps (Campos et al., 2024; Qi et al., 2023; Pan et al., 2023). Accurate predictions not only support better decision-making but also enhance system management and operational efficiency. While many existing forecasting methods primarily leverage historical values of the target series (i.e., endogenous variables), real-world applications often benefit from incorporating additional covariate series (i.e., exogenous variables). These covariates provide valuable supplementary information for predicting future endogenous values (Qiu et al., 2025e). Moreover, in many scenarios, future exogenous variables such as temperature, precipitation, and wind power can be reliably estimated using mature techniques, making them accessible for downstream prediction tasks. These variables are particularly valuable because their horizons align with those of the forecasting targets, offering direct predictive signals. In this work, we focus on forecasting with both historical and future exogenous variables, while also being able to accommodate scenarios where future exogenous variables are unavailable.

---

*Equal Contribution

[†]Corresponding Author

Temporal and channel correlations are prevalent in real-world scenarios (Yue et al., 2025; Huang et al.; Ma et al., 2025a; 2026), and accurately modeling them is crucial for effective forecasting with exogenous variables (Wang et al., 2024; Zhou et al., 2025). For instance, in electricity demand forecasting, historical electricity demand often exhibits similar temporal dependencies and periodicities to future electricity needs, demonstrating temporal correlations. If we consider temperature as a covariate in electricity forecasting, as shown in Figure 1, we may observe that higher temperatures tend to lead to higher electricity demand, which illustrates channel correlations. Many methods have been proposed to capture temporal and channel correlations, but most of them adopt a two-step modeling strategy, in which temporal correlations and channel correlations are modeled separately at different steps. As illustrated in Figure 2, these approaches either first model temporal correlations and then channel correlations (e.g., TimeXer (Wang et al., 2024), ExoTST (Tayal et al., 2024), Figure 2a), or first model channel correlations and then temporal correlations (e.g., TFT (Lim et al., 2021), CrossLinear (Zhou et al., 2025), Figure 2b). However, the two-step modeling strategy may lead to interference between the correlations learned in the two steps (Tu et al., 2025; Chi et al., 2025), which in turn limits the model's capacity to capture temporal and channel correlations, ultimately resulting in suboptimal performance. Graph-based approaches, however, are naturally well-suited for modeling correlations (Hu et al., 2025b; Chen et al., 2023; Shao et al., 2025; Liu et al., 2026a; Yu et al., 2025a). They can capture the relationships between nodes, enabling the joint modeling of temporal and channel correlations instead of modeling them separately.

Moreover, in real-world scenarios, sensor failures, transmission errors, and manual recording mistakes frequently occur, which introduce diverse types of noises into the data (Wang et al., 2024; Cheng et al., 2023). These issues further complicate the forecasting task, as the observed data may no longer accurately reflect the true correlations (Qiu et al., 2025a). As a result, conventional models tend to overfit noisy observations, which prevents them from capturing reliable correlations (Li et al., 2022; Wang et al., 2018). Generative models, however, have demonstrated strong performance in many time series analysis tasks (Li et al., 2022; Huang et al., 2022; Yu & Oh, 2021; DBL, 2022). They are designed to learn the underlying data distribution and capture the latent structures of time series, rather than inferring correlations directly from noisy observations, which can be misleading.

Inspired by the above observations, we propose **GCGNet**, a **G**raph-**C**onsistent **G**enerative **Net**work. The network aims to learn a graph that robustly captures the joint correlations across time and channels. Specifically, we first use a Variational Generator module to produce a coarse prediction. A Graph Structure Aligner module then guides the generator by evaluating the consistency between the generated and true correlations. The correlations are represented as graphs, produced by the Graph VAE module, and are robust to noises. Finally, a Graph Refiner module prevents potential degeneration in the Graph Structure Aligner module and improves the initial predictions by leveraging the learned correlations.

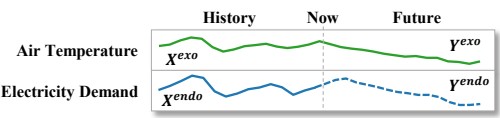

Figure 1: Illustration of forecasting Electricity Demand (i.e., the endogenous variable) with exogenous variable of Air Temperature. $X^{endo}$ and $Y^{endo}$ are historical and future endogenous variables, and $X^{exo}$ and $Y^{exo}$ are historical and future exogenous variables.

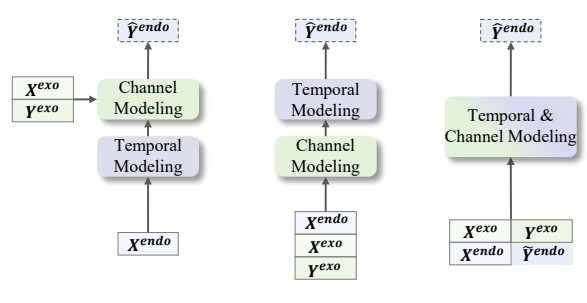

Figure 2: Categories of time series forecasting with exogenous variables algorithms: (a) modeling temporal correlations first, then modeling channel correlations; (b) modeling channel correlations first, then modeling temporal correlations; (c) jointly modeling temporal and channel correlations, where $\tilde{Y}^{endo}$ represents a coarse prediction of $Y^{endo}$ detailed in Section 3.2.

In summary, our contributions are as follows:

- We propose a general framework called GCGNet, which is designed for time series forecasting with historical and future exogenous variables in a robust manner. The framework achieves accurate forecasting through a graph-consistent generative network.

- We design the Graph Structure Aligner module, which leverages the consistency of graph structures to guide the generator, ensuring the outputs align with the underlying temporal and channel joint correlations and produce more accurate predictions.

- We conduct extensive experiments on 12 real-world datasets with exogenous variables. The results show that GCGNet outperforms state-of-the-art baselines. Additionally, all datasets and code are available at https://github.com/decisionintelligence/GCGNet.

## 2 RELATED WORK

### 2.1 TIME SERIES FORECASTING WITH EXOGENOUS VARIABLES

Time series forecasting with exogenous variables has been extensively studied in classical statistics. For example, ARIMAX and SARIMAX (Williams, 2001; Vagropoulos et al., 2016) extend ARIMA by incorporating correlations between exogenous and endogenous variables. Such methods rely on strong parametric assumptions regarding the functional form and distribution of the data, which may limit their flexibility when modeling complex real-world time series. In recent years, many deep learning methods have also been proposed for time series forecasting with exogenous variables. For example, TiDE (Das et al., 2023) adopts a coarse-grained strategy by directly concatenating exogenous variables with endogenous variables as inputs to the predictor. NBEATSx (Olivares et al., 2023) extends N-BEATS with dedicated branches to leverage exogenous signals. TFT (Lim et al., 2021) first applies variable selection to obtain step-wise representations and subsequently models temporal correlations. CrossLinear (Zhou et al., 2025) aggregates channel information via 1D convolutions before forecasting. TimeXer (Wang et al., 2024) models temporal dependencies first and then incorporates exogenous variables through cross-attention. Similarly, ExoTST (Tayal et al., 2024) also employs cross-attention but introduces additional designs for encoding exogenous variables. Most of these methods adopt a two-step modeling strategy to model temporal and channel correlations, which may result in mutual interference between the two steps. This separation may reduce the model's ability to fully capture temporal and channel correlations.

### 2.2 GENERATIVE MODELS FOR TIME SERIES

Generative models have been widely applied to various time series tasks due to their ability to capture underlying distributions and latent structures. For generation, models such as TimeGAN (Yoon et al., 2019) and GT-GAN (DBL, 2022) produce realistic sequences by jointly optimizing supervised and adversarial objectives, while VAE-based methods (Desai et al., 2023; Wu et al., 2025b) learn latent representations that preserve the statistical properties of the data. For imputation, generative models recover missing values by capturing the latent structures (Yu & Oh, 2021; Boquet et al., 2019). In anomaly detection, variational methods capture normal temporal patterns and identify unexpected patterns as anomalies (Huang et al., 2022; Wang et al., 2023a; Kieu et al., 2022), while GAN-based methods detect anomalies by contrasting generated and observed sequences (Li et al., 2019). For forecasting, D3VAE (Li et al., 2022) combines variational autoencoders with diffusion-based denoising to improve the accuracy of time series prediction. In our work, we leverage generative models for time series forecasting with exogenous variables, enabling robust modeling of temporal and channel correlations.

## 3 METHODOLOGY

In the context of exogenous-aware time series forecasting, the historical endogenous time series $X^{\text{endo}} \in \mathbb{R}^{N \times T}$ is accompanied by historical exogenous variables $X^{\text{exo}} \in \mathbb{R}^{D \times T}$ and future exogenous variables $Y^{\text{exo}} \in \mathbb{R}^{D \times F}$, where $N$ is the number of endogenous variables, $D$ is the number of exogenous variables, $T$ is the number of historical time steps, and $F$ is the forecasting horizon. The objective is to forecast the future endogenous series $\hat{Y}^{\text{endo}} \in \mathbb{R}^{N \times F}$ over the horizon $F$ based on the

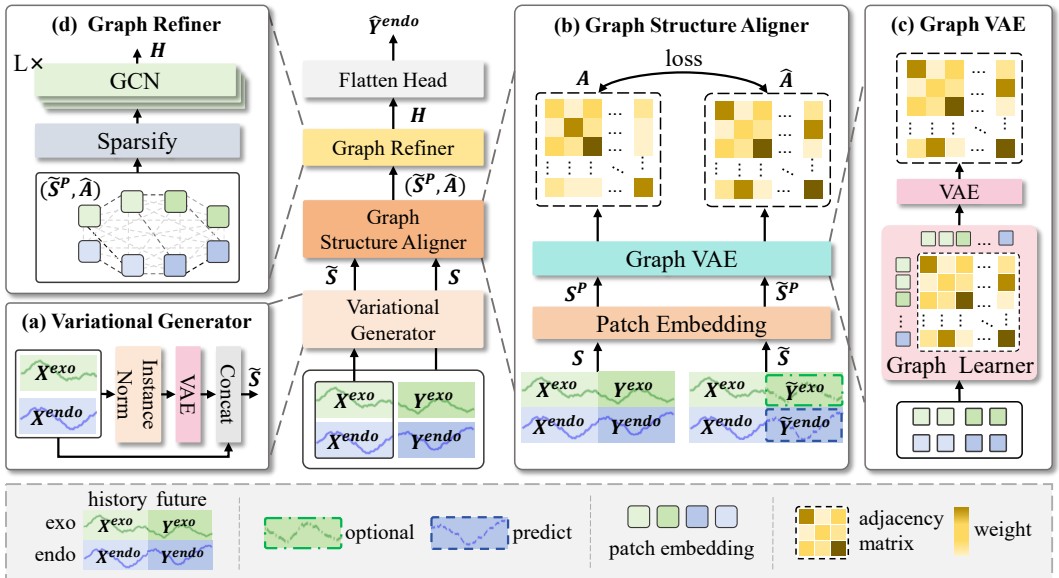

Figure 3: The architecture of GCGNet. (a) The Variational Generator, which produces coarse prediction. (b) The Graph Structure Aligner, which guides the optimization of the Variational Generator. (c) The Graph VAE, which generates the adjacency matrix for the nodes. (d) The Graph Refiner, which refines the final predictions using the learned adjacency matrix.

given historical endogenous and exogenous series, which can be formally expressed as follows.

$$\hat{Y}^{\text{endo}} = \mathcal{F}_\theta(X^{\text{endo}}, X^{\text{exo}}, Y^{\text{exo}}) \tag{1}$$

## 3.1 STRUCTURE OVERVIEW

In this work, we aim to model the joint temporal and channel correlations in a robust manner by integrating graph-based and generative models. Figure 3 shows the overall architecture of **GCGNet**, which consists of three main modules: the Variational Generator module, the Graph Structure Aligner module and the Graph Refiner module. First, we introduce the Variational Generator module to generate a coarse prediction, which provides an initial estimate for subsequent modules to guide optimization and correlations modeling. Next, we adopt the Graph Structure Aligner module to guide the optimization of the Variational Generator module. In addition, it generates an adjacency matrix $\hat{A}$, which captures joint correlations over time and channels. Specifically, a Graph VAE is utilized within the Graph Structure Aligner module to generate the adjacency matrix $\hat{A}$, which robustly represents these correlations. Finally, we employ the Graph Refiner module to enhance the initial predictions and prevent the degeneration (i.e., the module produces identical outputs regardless of the input) of the Graph Structure Aligner module. By leveraging $\hat{A}$, the Graph Refiner module facilitates information exchange across both temporal steps and channels. After refinement, a flatten prediction head is applied to produce the final forecast $\hat{Y}^{\text{endo}}$. We further introduce the details of each module in the following sections.

## 3.2 VARIATIONAL GENERATOR

The Variational Generator module aims to produce a coarse generation of the future sequence, facilitating the subsequent modeling of dependencies. Specifically, we first apply the Instance Normalization (Ulyanov et al., 2016) to unify the distributions of the training and testing data. Subsequently, the historical endogenous and exogenous variables $X^{\text{endo}}$ and $X^{\text{exo}}$ are fed into a VAE to generate the coarse predictions $\tilde{Y}^{\text{endo}}$ and $\tilde{Y}^{\text{exo}}$. Note that the generation of $\tilde{Y}^{\text{exo}}$ is optional: when the future exogenous variables are available, the real $Y^{\text{exo}}$ is directly used instead of the generated one. This design is motivated by the fact that future exogenous variables are not always accessible in real-world scenarios. The generated results are then concatenated with $X^{\text{endo}}$ and $X^{\text{exo}}$ to construct the

input for the Graph Structure Aligner module. The process can be formalized as follows.

$$\tilde{Y}^{\text{endo}} = \text{VAE}(X^{\text{endo}}), \quad \tilde{Y}^{\text{exo}} = \text{VAE}(X^{\text{exo}}) \tag{2}$$

$$\tilde{S} = \begin{pmatrix} X^{\text{exo}} & Z \\ X^{\text{endo}} & \tilde{Y}^{\text{endo}} \end{pmatrix} \in \mathbb{R}^{(N+D)\times(T+F)}, \quad Z = \begin{cases} Y^{\text{exo}}, & \text{if available} \\ \tilde{Y}^{\text{exo}}, & \text{otherwise} \end{cases} \tag{3}$$

where $\tilde{S}$ denotes the generated full sequence. $Z$ takes the observed $Y^{\text{exo}}$ if future exogenous variables exist, otherwise the predicted $\tilde{Y}^{\text{exo}}$. $S = \begin{pmatrix} X^{\text{exo}} & Y^{\text{exo}} \\ X^{\text{endo}} & Y^{\text{endo}} \end{pmatrix} \in \mathbb{R}^{(N+D)\times(T+F)}$ denotes the ground-truth full sequence. Both $\tilde{S}$ and $S$ are then fed into the Graph Structure Aligner module.

## 3.3 GRAPH STRUCTURE ALIGNER

The Graph Structure Aligner module is introduced to constrain the generative process of the Variational Generator module through the alignment of correlations. As shown in Figure 3b, rather than relying solely on point-wise reconstruction losses between predicted and ground-truth series, the aligner enforces consistency at the graph level, which encourages the generator to better focus on the underlying temporal and variable dependencies. Specifically, given the generated full sequence $\tilde{S}$ and the ground-truth full sequence $S$, we first obtain patch-wise representations to better model the time series (Cirstea et al., 2022; Nie et al., 2023; Wu et al., 2025a; Huang et al., 2024).

$$S^p = \text{Embedding}(\text{Patchify}(S)), \tilde{S}^p = \text{Embedding}(\text{Patchify}(\tilde{S})), \tag{4}$$

where $S^p, \tilde{S}^p \in \mathbb{R}^{(N+D)\times L\times d}$. Specifically, Patchify module first segments $S$ and $\tilde{S}$ along the temporal dimension into non-overlapping patches with patch size $p$, resulting in $L = \lceil (T+F)/p \rceil$ patches. Then, the embedding layer maps each patch of length $p$ into a $d$-dimensional vector representation.

**Graph VAE.** As shown in Figure 3c, to capture the relational structures among the patches, we feed the patch embeddings into the Graph VAE. For illustration, consider an example input $X^p \in \mathbb{R}^{(N+D)\times L\times d}$, where in practice the input can be $S^p$ or $\tilde{S}^p$. The adjacency computation is performed as follows.

$$A' = \text{GELU}\big((W_1 X^p)(W_2 X^p)^\top\big) \tag{5}$$

$$\tilde{A} = \frac{1}{2}\big(A' + A'^\top\big), \tag{6}$$

where $A', \tilde{A} \in \mathbb{R}^{L\times L}$, $W_1, W_2 \in \mathbb{R}^{d\times d}$ are learnable projection matrices. The symmetrization step, $\tilde{A} = \frac{1}{2}(A' + A'^\top)$, ensures that the resulting graph is undirected by making the pairwise relationships consistent. While $\tilde{A}$ encodes pairwise joint correlations among patches, it may contain noises and fail to capture higher-order dependencies. To obtain more robust relational structures, we further employ a VAE (Simonovsky & Komodakis, 2018) module, which not only denoises the raw similarity graph but also introduces a probabilistic latent space to model uncertainty in the learned adjacency. By providing a smoothed graph representation, the VAE enables the Graph Structure Aligner module to align structures more reliably, improving stability and generalization.

$$A = VAE(\tilde{A}) \tag{7}$$

Finally, the aligner enforces structural alignment by minimizing the difference between $A$ and $\hat{A}$. Here, $A$ captures the joint correlations among the ground-truth full sequence $S$, while $\hat{A}$ captures the correlations among the generated full sequence $\tilde{S}$. Formally, the loss is defined as follows.

$$\mathcal{L}_{align} = \|A - \hat{A}\|_1, \tag{8}$$

where $\|\cdot\|_1$ denotes the L1 distance over elements. Optimizing this loss encourages the generator to produce sequences whose patch-wise relationships are consistent with the ground-truth structures, thereby considering temporal and channel correlations.

### 3.4 GRAPH REFINER

In the Graph Structure Aligner module, both $A$ and $\hat{A}$ are obtained from a shared Graph VAE. This setup, however, can lead to degeneration, where the Graph VAE produces identical outputs regardless of the input. To mitigate this issue, the Graph Refiner module is introduced. By using $\hat{A}$ to refine the predictions, the Graph Refiner module forces the Graph VAE to produce meaningful adjacency matrices, as any degeneration would directly increase the prediction loss. At the same time, the Graph Refiner module integrates dependencies across both temporal steps and channels, facilitating information exchange and resulting in more accurate final predictions.

Specifically, the Graph Refiner module takes two inputs: the generated sequence patch embeddings $\tilde{S}^p$, and the adjacency matrix $\hat{A}$ produced by the Graph Structure Aligner module. Here, $\tilde{S}^p$ serves as the nodes of the graph, and $\hat{A}$ serves as the edges. Directly utilizing the potentially dense matrix $\hat{A}$ may introduce noises from insignificant correlations. To retain only the most significant connections, we first apply a Sparsify step. We adopt a top-$k$ selection strategy following TimeFilter (Hu et al., 2025b). For each node, only the edges with the $k$ largest weights are preserved, resulting in a sparse adjacency matrix $A_s$.

This Sparsify guides the model to focus on the significant connections across both temporal steps and channels. With the sparse graph structures $A_s$ and the initial node features $\tilde{S}^p$, we apply a multi-layer Graph Convolutional Network (GCN) to propagate information. By stacking multiple GCN layers, each node can aggregate information from its neighborhoods, enabling the Graph Refiner module to capture both local and global dependencies and refine feature representations.

$$H = GCN(\tilde{S}^p, A_s)), \tag{9}$$

where $H \in \mathbb{R}^{(N+D) \times L \times d}$ shares the same shape as $\tilde{S}^p$. Finally, the refined representations are flattened and projected through a linear layer to obtain the final output $\hat{Y}^{\text{endo}} \in \mathbb{R}^{N \times F}$.

$$\hat{Y}^{endo} = \text{Linear}(\text{Flatten}(H)) \tag{10}$$

### 3.5 LOSS FUNCTION

The training objective of GCGNet consists of four components: 1) the *aligner loss* $L_{align}$ introduced in Equation 8 in Section 3.3, which enforces structural alignment between generated and ground-truth full sequences, guiding the optimization of the Variational Generator module. 2) the *forecasting loss* $L_f$, which directly supervises the final prediction of future endogenous variables produced by the Graph Refiner module. 3) the *variational regularization loss* $L_{KL}^V$ from the VAE in the Variational Generator module and 4) the *graph regularization loss* $L_{KL}^G$ from the Graph VAE. The KL terms regularize the latent space of their respective generators.

The forecasting loss is defined as follows.

$$L_f = \left\| Y^{endo} - \hat{Y}^{endo} \right\|_1 \tag{11}$$

The variational regularization terms are defined as follows.

$$L_{KL}^V = D_{KL}\big(q(Z^V|X) \,\|\, p(Z^V)\big), \quad L_{KL}^G = D_{KL}\big(q(Z^{\hat{A}}|\hat{A}) \,\|\, p(Z^{\hat{A}})\big), \tag{12}$$

where $q(\cdot)$ and $p(\cdot)$ denote the approximate posterior and prior distributions, $Z^V$ denotes the latent variable in the Variational Generator VAE, and $Z^{\hat{A}}$ denotes the latent variable in the Graph VAE used within the Graph Structure Aligner module.

Finally, the overall loss function is defined as the sum of all components.

$$L_{total} = L_f + L_{align} + L_{KL}^V + L_{KL}^G \tag{13}$$

## 4 EXPERIMENTS

**Datasets** We conduct experiments on 12 real-world datasets with exogenous variables to comprehensively evaluate the performance of GCGNet. In these datasets, the future exogenous variables

Table 1: Mean Absolute Error (MAE) and Mean Squared Error (MSE) on 12 real-world datasets with exogenous variables. The best results are **Red**, and the second-best results are Blue. Avg represents the average results across the two forecasting horizons.

| Models | | GCGNet | | TimeXer | | TFT | | TiDE | | DUET | | CrossLinear | | Amplifier | | TimeKAN | | xPatch | | PatchTST | |
|---|---|---|---|---|---|---|---|---|---|---|---|---|---|---|---|---|---|---|---|---|---|
| Metrics | | mse | mae | mse | mae | mse | mae | mse | mae | mse | mae | mse | mae | mse | mae | mse | mae | mse | mae | mse | mae |
| NP | 24 | **0.197** | **0.233** | 0.236 | 0.266 | 0.219 | 0.249 | 0.284 | 0.301 | 0.246 | 0.287 | 0.215 | 0.261 | 0.252 | 0.303 | 0.273 | 0.310 | 0.234 | 0.278 | 0.249 | 0.294 |
| | 360 | **0.496** | **0.442** | 0.600 | 0.475 | 0.539 | 0.501 | 0.601 | 0.498 | 0.576 | 0.528 | 0.593 | 0.508 | 0.587 | 0.534 | 0.538 | 0.529 | 0.523 | 0.461 | 0.530 | 0.498 |
| | Avg | **0.346** | **0.337** | 0.418 | 0.371 | 0.379 | 0.375 | 0.443 | 0.400 | 0.411 | 0.408 | 0.420 | 0.384 | 0.420 | 0.419 | 0.405 | 0.419 | 0.378 | 0.370 | 0.390 | 0.396 |
| PJM | 24 | **0.058** | **0.148** | 0.075 | 0.166 | 0.095 | 0.195 | 0.106 | 0.214 | 0.072 | 0.166 | 0.094 | 0.197 | 0.096 | 0.208 | 0.115 | 0.244 | 0.077 | 0.168 | 0.116 | 0.239 |
| | 360 | **0.128** | 0.223 | 0.140 | 0.231 | 0.133 | **0.219** | 0.177 | 0.279 | 0.131 | 0.228 | 0.142 | 0.254 | 0.177 | 0.285 | 0.162 | 0.281 | 0.131 | 0.220 | 0.149 | 0.279 |
| | Avg | **0.093** | **0.186** | 0.108 | 0.198 | 0.114 | 0.207 | 0.142 | 0.246 | 0.102 | 0.197 | 0.118 | 0.226 | 0.137 | 0.246 | 0.139 | 0.262 | 0.104 | 0.194 | 0.133 | 0.259 |
| BE | 24 | **0.323** | **0.227** | 0.392 | 0.253 | 0.426 | 0.272 | 0.426 | 0.285 | 0.432 | 0.272 | 0.383 | 0.251 | 0.471 | 0.339 | 0.451 | 0.319 | 0.411 | 0.256 | 0.452 | 0.326 |
| | 360 | 0.524 | 0.347 | 0.512 | 0.327 | **0.482** | **0.310** | 0.571 | 0.364 | 0.597 | 0.452 | 0.564 | 0.408 | 0.646 | 0.487 | 0.645 | 0.495 | 0.579 | 0.410 | 0.702 | 0.538 |
| | Avg | **0.423** | **0.287** | 0.452 | 0.290 | 0.454 | 0.291 | 0.498 | 0.325 | 0.515 | 0.354 | 0.474 | 0.330 | 0.559 | 0.413 | 0.548 | 0.407 | 0.495 | 0.333 | 0.577 | 0.432 |
| FR | 24 | **0.332** | **0.179** | 0.366 | 0.208 | 0.543 | 0.253 | 0.418 | 0.255 | 0.384 | 0.251 | 0.405 | 0.228 | 0.459 | 0.348 | 0.454 | 0.296 | 0.424 | 0.235 | 0.518 | 0.368 |
| | 360 | 0.478 | 0.280 | 0.489 | 0.273 | **0.465** | **0.261** | 0.551 | 0.308 | 0.607 | 0.403 | 0.567 | 0.371 | 0.648 | 0.468 | 0.641 | 0.452 | 0.556 | 0.350 | 0.658 | 0.452 |
| | Avg | **0.405** | **0.230** | 0.427 | 0.241 | 0.504 | 0.257 | 0.484 | 0.281 | 0.496 | 0.327 | 0.486 | 0.300 | 0.554 | 0.408 | 0.547 | 0.374 | 0.490 | 0.293 | 0.588 | 0.410 |
| DE | 24 | **0.282** | **0.328** | 0.339 | 0.362 | 0.380 | 0.383 | 0.367 | 0.383 | 0.376 | 0.378 | 0.384 | 0.390 | 0.394 | 0.407 | 0.399 | 0.412 | 0.399 | 0.394 | 0.413 | 0.412 |
| | 360 | **0.491** | **0.445** | 0.610 | 0.474 | 0.599 | 0.509 | 0.630 | 0.511 | 0.589 | 0.482 | 0.610 | 0.516 | 0.551 | 0.474 | 0.547 | 0.479 | 0.550 | 0.472 | 0.589 | 0.498 |
| | Avg | **0.387** | **0.387** | 0.475 | 0.418 | 0.489 | 0.446 | 0.499 | 0.447 | 0.482 | 0.430 | 0.497 | 0.453 | 0.473 | 0.441 | 0.473 | 0.445 | 0.474 | 0.433 | 0.501 | 0.455 |
| Energy | 24 | **0.065** | **0.194** | 0.122 | 0.273 | 0.093 | 0.235 | 0.103 | 0.248 | 0.117 | 0.283 | 0.093 | 0.241 | 0.138 | 0.306 | 0.135 | 0.298 | 0.092 | 0.240 | 0.239 | 0.390 |
| | 360 | **0.179** | **0.330** | 0.204 | 0.357 | 0.167 | 0.331 | 0.202 | 0.355 | 0.288 | 0.452 | 0.273 | 0.406 | 0.328 | 0.472 | 0.302 | 0.464 | 0.204 | 0.367 | 0.214 | 0.363 |
| | Avg | **0.122** | **0.262** | 0.163 | 0.315 | 0.130 | 0.283 | 0.153 | 0.302 | 0.203 | 0.367 | 0.183 | 0.324 | 0.233 | 0.389 | 0.218 | 0.381 | 0.148 | 0.303 | 0.226 | 0.377 |
| Sdwpfm1 | 24 | 0.375 | **0.409** | 0.558 | 0.533 | 0.366 | 0.421 | 0.474 | 0.488 | 0.551 | 0.564 | **0.345** | 0.433 | 0.364 | 0.445 | 0.418 | 0.503 | 0.669 | 0.580 | 0.374 | 0.454 |
| | 360 | **0.458** | **0.490** | 0.845 | 0.684 | 0.597 | 0.528 | 0.492 | 0.526 | 0.646 | 0.577 | 0.488 | 0.523 | 0.510 | 0.535 | 0.476 | 0.566 | 0.592 | 0.556 | 0.497 | 0.550 |
| | Avg | **0.416** | **0.449** | 0.701 | 0.609 | 0.482 | 0.474 | 0.483 | 0.507 | 0.599 | 0.570 | 0.417 | 0.478 | 0.437 | 0.490 | 0.447 | 0.534 | 0.630 | 0.568 | 0.435 | 0.502 |
| Sdwpfm2 | 24 | 0.410 | 0.443 | 0.627 | 0.570 | 0.411 | 0.458 | 0.461 | 0.492 | 0.445 | 0.452 | 0.570 | 0.563 | **0.394** | 0.462 | 0.474 | 0.538 | 0.508 | 0.478 | 0.418 | 0.492 |
| | 360 | **0.505** | **0.514** | 0.978 | 0.736 | 0.541 | 0.519 | 0.511 | 0.540 | 0.584 | 0.528 | 0.755 | 0.666 | 0.587 | 0.563 | 0.520 | 0.589 | 0.523 | 0.513 | 0.602 | 0.603 |
| | Avg | **0.458** | **0.479** | 0.803 | 0.653 | 0.476 | 0.488 | 0.486 | 0.516 | 0.514 | 0.490 | 0.662 | 0.615 | 0.491 | 0.512 | 0.497 | 0.564 | 0.516 | 0.495 | 0.510 | 0.547 |
| Sdwpfh1 | 24 | **0.381** | **0.441** | 0.651 | 0.587 | 0.401 | 0.460 | 0.434 | 0.489 | 0.527 | 0.513 | 0.614 | 0.601 | 0.576 | 0.627 | 0.511 | 0.582 | 0.524 | 0.505 | 0.422 | 0.505 |
| | 360 | **0.446** | **0.495** | 0.841 | 0.700 | 0.557 | 0.523 | 0.472 | 0.527 | 0.551 | 0.519 | 0.728 | 0.650 | 0.497 | 0.569 | 0.643 | 0.694 | 0.472 | 0.504 | 0.513 | 0.548 |
| | Avg | **0.414** | **0.468** | 0.746 | 0.643 | 0.479 | 0.491 | 0.453 | 0.508 | 0.539 | 0.516 | 0.671 | 0.625 | 0.537 | 0.598 | 0.577 | 0.638 | 0.498 | 0.505 | 0.468 | 0.527 |
| Sdwpfh2 | 24 | **0.426** | **0.474** | 0.820 | 0.677 | 0.474 | 0.493 | 0.579 | 0.553 | 0.629 | 0.563 | 0.585 | 0.586 | 0.473 | 0.533 | 0.580 | 0.614 | 0.640 | 0.554 | 0.457 | 0.527 |
| | 360 | **0.519** | **0.529** | 0.962 | 0.761 | 0.657 | 0.549 | 0.619 | 0.614 | 0.665 | 0.569 | 0.656 | 0.605 | 0.569 | 0.628 | 0.713 | 0.729 | 0.579 | 0.547 | 0.641 | 0.632 |
| | Avg | **0.472** | **0.501** | 0.891 | 0.719 | 0.566 | 0.521 | 0.599 | 0.583 | 0.647 | 0.566 | 0.647 | 0.566 | 0.521 | 0.581 | 0.647 | 0.672 | 0.610 | 0.551 | 0.549 | 0.580 |
| Colbun | 10 | **0.057** | **0.084** | 0.113 | 0.172 | 0.092 | 0.135 | 0.089 | 0.131 | 0.089 | 0.134 | 0.063 | 0.107 | 0.071 | 0.121 | 0.061 | 0.101 | 0.090 | 0.104 | 0.062 | 0.122 |
| | 30 | **0.138** | **0.224** | 0.176 | 0.299 | 0.383 | 0.460 | 0.240 | 0.322 | 0.307 | 0.397 | 0.300 | 0.265 | 0.275 | 0.370 | 0.195 | 0.249 | 0.217 | 0.315 | 0.417 | 0.496 |
| | Avg | **0.098** | **0.154** | 0.145 | 0.235 | 0.238 | 0.297 | 0.164 | 0.227 | 0.198 | 0.266 | 0.181 | 0.186 | 0.173 | 0.246 | 0.128 | 0.175 | 0.153 | 0.210 | 0.239 | 0.309 |
| Rapel | 10 | **0.140** | **0.191** | 0.301 | 0.308 | 0.201 | 0.253 | 0.228 | 0.271 | 0.174 | 0.219 | 0.271 | 0.248 | 0.181 | 0.227 | 0.174 | 0.231 | 0.161 | 0.216 | 0.163 | 0.218 |
| | 30 | **0.284** | **0.327** | 0.387 | 0.416 | 0.409 | 0.414 | 0.411 | 0.432 | 0.365 | 0.432 | 0.271 | 0.371 | 0.333 | 0.390 | 0.325 | 0.390 | 0.505 | 0.546 | 0.374 | 0.445 |
| | Avg | **0.212** | **0.259** | 0.344 | 0.362 | 0.305 | 0.333 | 0.320 | 0.351 | 0.269 | 0.326 | 0.271 | 0.310 | 0.257 | 0.321 | 0.249 | 0.311 | 0.333 | 0.381 | 0.269 | 0.332 |
| 1st Count | | **30** | **32** | 0 | 0 | 3 | 3 | 0 | 0 | 0 | 0 | 2 | 0 | 1 | 0 | 0 | 0 | 0 | 1 | 0 | 0 |

are available. We also consider settings where future exogenous variables are unavailable (see Section 4.2). Further details of the datasets are provided in Appendix A.1.

**Baselines** We conduct a comprehensive comparison against 10 baseline models, which include 1) methods that inherently support future exogenous variables, such as TimeXer (Wang et al., 2024), TFT (Lim et al., 2021), and TiDE (Das et al., 2023); and 2) methods that do not inherently support future exogenous variables, including DUET (Qiu et al., 2025d), CrossLinear (Zhou et al., 2025), Amplifier (Fei et al., 2025), TimeKAN (Huang et al., 2025b), xPatch (Stitsyuk & Choi, 2025), and PatchTST (Nie et al., 2023). For the latter group, we extend their architectures by integrating future exogenous variables through an MLP fusion module, as detailed in Appendix A.3. To ensure fairness, all models are evaluated using the same inputs, namely historical endogenous variables, historical exogenous variables, and future exogenous variables.

**Setup** To ensure consistency with previous studies, we adopt Mean Squared Error (MSE) and Mean Absolute Error (MAE) as evaluation metrics. We conduct both short-term and long-term prediction experiments across all datasets. For the Colbun and Rapel datasets, the short-term setting uses a lookback window of 60 with a prediction horizon of 10, while the long-term setting uses a lookback window of 180 with a prediction horizon of 30. For the other datasets, the short-term setting uses a lookback window of 168 with a prediction horizon of 24, and the long-term setting uses a lookback window of 720 with a prediction horizon of 360.

## 4.1 MAIN RESULTS

Comprehensive forecasting results are presented in Table 1 to demonstrate the performance of GCGNet on the datasets with exogenous variables. Across both long-term and short-term fore-

Table 2: Ablation studies for GCGNet. The best results are highlighted in **bold**.

| Dataset | NP | | PJM | | DE | | Energy | | Average | |
|---|---|---|---|---|---|---|---|---|---|---|
| Metrics | mse | mae | mse | mae | mse | mae | mse | mae | mse | mae |
| (a) Replace Variational Generator | 0.537 | 0.464 | 0.140 | 0.237 | 0.548 | 0.471 | 0.226 | 0.374 | 0.363 | 0.386 |
| (b) Remove $L_{align}$ | 0.659 | 0.526 | 0.137 | 0.230 | 0.791 | 0.584 | 0.308 | 0.446 | 0.474 | 0.446 |
| (c) Replace Graph VAE | 0.691 | 0.540 | 0.213 | 0.290 | 0.665 | 0.526 | 0.228 | 0.376 | 0.449 | 0.433 |
| (d) Remove Graph Refiner | 0.853 | 0.599 | 0.320 | 0.387 | 0.970 | 0.653 | 0.278 | 0.413 | 0.605 | 0.513 |
| GCGNet | **0.496** | **0.442** | **0.128** | **0.223** | **0.491** | **0.445** | **0.179** | **0.330** | **0.323** | **0.360** |

casting settings on 12 real-world datasets, GCGNet achieves outstanding performance compared to forecasters of various architectures. Specifically, as shown in Table 1, GCGNet achieves 18 first-place rankings in MSE and 20 in MAE. The performance demonstrates its clear superiority over models specifically designed for time series forecasting with exogenous variables that adopt a two-step modeling strategy, such as TimeXer and TFT, as well as recent state-of-the-art forecasting models adapted to this setting, such as DUET and Amplifier. Moreover, it can also be observed that models originally designed to incorporate both historical and future exogenous variables, such as TimeXer and TFT, exhibit relatively strong performance, highlighting the importance of the design of leveraging future exogenous variables in this setting.

## 4.2 MODEL ANALYSES

**Ablation studies** To ascertain the impact of different modules within GCGNet, we perform ablation studies focusing on the following components: (a) Replacing the VAE with an MLP in the Variational Generator module. (b) Removing the $L_{align}$ in the Graph Structure Aligner module, while still using $\tilde{S}$ to obtain $(\tilde{S}^p, \hat{a})$. (c) Replacing the Graph VAE with a Graph Learner, directly adopting the adjacency matrix produced by the Graph Learner in the Graph VAE. (d) Removing the entire Graph Refiner module, so that the model output is directly generated by the Variational Generator. The prediction remains under the guidance of the Graph Structure Aligner module and the loss is still computed against the ground-truth $Y^{endo}$. We have the following observations: 1) Replacing the VAE with an MLP degrades performance, which demonstrates the necessity of variational modeling in capturing the uncertainty and diversity of data. 2) When the $L_{align}$ term is removed, the performance decreases because the Graph Structure Aligner module no longer provides structural guidance. This indicates that $L_{align}$ is crucial for encouraging the Variational Generator module to produce coarse forecasts that are consistent with the underlying correlations. 3) Replacing the Graph VAE with a Graph Learner, which directly adopts the adjacency matrix produced by the Graph Learner, performs worse. Unlike the Graph VAE, the Graph Learner generates deterministic graphs and cannot capture the uncertainty and diversity of possible graph structures. Consequently, the learned graphs are less expressive and less robust, limiting the model's ability to represent complex temporal and channel correlations. 4) Removing the Graph Refiner module results in a significant drop in performance, since the Graph Structure Aligner module is applied but its generated adjacency matrix is not used, the Graph VAE tends to degenerate into producing meaningless outputs, which in turn causes degeneration. The Graph Refiner module prevents this by incorporating $\tilde{A}$ into the prediction process, thereby forcing the VAE to produce informative adjacency matrices and enabling the model to capture temporal and channel correlations.

**Forecasting without Future Exogenous Variables** Considering that future exogenous variables may not always be available in practice, we conduct experiments using only historical endogenous and exogenous variables to assess the robustness and applicability of GCGNet (see Table 3). When future exogenous variables $Y^{exo}$ are unavailable, GCGNet replaces them with predictions $\tilde{Y}^{exo}$ from the Variational Generator module, while keeping the rest of the pipeline unchanged. For all baseline models, we consistently provide only historical variables as input. Methods such as TimeXer and CrossLinear are capable of working without future exogenous variables. DUET, which models channel dependencies, also supports historical endogenous and exogenous inputs. In contrast, the channel-independent model PatchTST uses only historical endogenous variables as input. The experimental results show that GCGNet continues to perform excellently. Methods that use historical exogenous variables, such as TimeXer and CrossLinear, also perform well. DUET, which flexibly

Table 3: Average results under the setting where future exogenous variables are not available. The inputs are ($X^{\text{endo}}$ and $X^{\text{exo}}$). The best results are Red, and the second-best results are Blue. Full results are provided in Table 7 of Appendix B.2.

| Models | GCGNet | | TimeXer | | TFT | | TiDE | | DUET | | CrossLinear | | Amplifier | | TimeKAN | | xPatch | | PatchTST | |
|---|---|---|---|---|---|---|---|---|---|---|---|---|---|---|---|---|---|---|---|---|
| Metrics | mse | mae | mse | mae | mse | mae | mse | mae | mse | mae | mse | mae | mse | mae | mse | mae | mse | mae | mse | mae |
| NP | 0.425 | 0.377 | 0.440 | 0.383 | 0.647 | 0.488 | 0.467 | 0.416 | 0.444 | 0.383 | 0.451 | 0.394 | 0.520 | 0.448 | 0.484 | 0.435 | 0.488 | 0.402 | 0.457 | 0.401 |
| PJM | 0.133 | 0.217 | 0.141 | 0.229 | 0.200 | 0.270 | 0.158 | 0.253 | 0.140 | 0.226 | 0.147 | 0.241 | 0.152 | 0.245 | 0.175 | 0.270 | 0.139 | 0.223 | 0.148 | 0.243 |
| BE | 0.458 | 0.301 | 0.477 | 0.301 | 0.563 | 0.354 | 0.547 | 0.348 | 0.473 | 0.305 | 0.477 | 0.300 | 0.502 | 0.325 | 0.488 | 0.315 | 0.483 | 0.302 | 0.485 | 0.309 |
| FR | 0.428 | 0.245 | 0.454 | 0.247 | 0.535 | 0.288 | 0.494 | 0.290 | 0.468 | 0.262 | 0.476 | 0.257 | 0.494 | 0.286 | 0.491 | 0.276 | 0.477 | 0.257 | 0.470 | 0.264 |
| DE | 0.611 | 0.497 | 0.659 | 0.507 | 0.684 | 0.515 | 0.644 | 0.519 | 0.660 | 0.513 | 0.635 | 0.508 | 0.712 | 0.548 | 0.669 | 0.526 | 0.633 | 0.500 | 0.696 | 0.527 |
| Energy | 0.150 | 0.299 | 0.172 | 0.326 | 0.376 | 0.480 | 0.162 | 0.311 | 0.160 | 0.308 | 0.201 | 0.336 | 0.180 | 0.327 | 0.147 | 0.296 | 0.170 | 0.314 | 0.203 | 0.341 |
| Sdwpfm1 | 0.700 | 0.602 | 0.701 | 0.609 | 0.984 | 0.728 | 0.713 | 0.633 | 0.724 | 0.583 | 0.843 | 0.685 | 0.725 | 0.672 | 0.737 | 0.651 | 0.733 | 0.651 | 0.733 | 0.651 |
| Sdwpfm2 | 0.825 | 0.633 | 0.803 | 0.653 | 1.044 | 0.767 | 0.816 | 0.682 | 0.820 | 0.641 | 0.800 | 0.687 | 0.942 | 0.732 | 0.836 | 0.701 | 0.850 | 0.632 | 0.834 | 0.714 |
| Sdwpfh1 | 0.757 | 0.628 | 0.746 | 0.643 | 0.793 | 0.698 | 0.808 | 0.699 | 0.780 | 0.637 | 0.768 | 0.702 | 1.100 | 0.820 | 0.804 | 0.720 | 0.791 | 0.630 | 0.804 | 0.717 |
| Sdwpfh2 | 0.882 | 0.691 | 0.891 | 0.719 | 0.926 | 0.746 | 0.919 | 0.751 | 0.920 | 0.691 | 0.956 | 0.774 | 0.980 | 0.768 | 0.941 | 0.782 | 0.941 | 0.687 | 1.025 | 0.802 |
| Colbun | 0.116 | 0.164 | 0.132 | 0.219 | 0.556 | 0.386 | 0.188 | 0.240 | 0.147 | 0.198 | 0.129 | 0.195 | 0.152 | 0.210 | 0.201 | 0.253 | 0.238 | 0.235 | 0.150 | 0.221 |
| Rapel | 0.258 | 0.273 | 0.344 | 0.362 | 0.308 | 0.334 | 0.323 | 0.357 | 0.304 | 0.306 | 0.240 | 0.299 | 0.337 | 0.348 | 0.342 | 0.337 | 0.364 | 0.382 | 0.325 | 0.334 |
| 1st Count | 8 | 7 | 1 | 0 | 0 | 0 | 0 | 0 | 0 | 1 | 2 | 1 | 0 | 0 | 1 | 1 | 0 | 2 | 0 | 0 |

Table 4: Average results under the setting where exogenous variables are partial missing. The best results are highlighted in **bold**. Full results are provided in Table 10 of Appendix B.4.

| Mask Type | Zero | | | | | | | | | | Random | | | | | | | | | |
|---|---|---|---|---|---|---|---|---|---|---|---|---|---|---|---|---|---|---|---|---|
| Models | GCGNet | | TimeXer | | CrossLinear | | DUET | | PatchTST | | GCGNet | | TimeXer | | CrossLinear | | DUET | | PatchTST | |
| Metrics | mse | mae | mse | mae | mse | mae | mse | mae | mse | mae | mse | mae | mse | mae | mse | mae | mse | mae | mse | mae |
| Colbun | 0.076 | **0.093** | 0.084 | 0.145 | **0.065** | 0.102 | 0.106 | 0.126 | 0.070 | 0.108 | 0.070 | **0.096** | 0.112 | 0.169 | **0.066** | 0.108 | 0.084 | 0.118 | 0.078 | 0.120 |
| Energy | **0.070** | **0.203** | 0.122 | 0.273 | 0.097 | 0.241 | 0.100 | 0.244 | 0.108 | 0.257 | **0.081** | **0.217** | 0.123 | 0.275 | 0.098 | 0.242 | 0.100 | 0.245 | 0.102 | 0.249 |
| DE | **0.292** | **0.336** | 0.418 | 0.403 | 0.426 | 0.416 | 0.428 | 0.407 | 0.480 | 0.445 | **0.344** | **0.366** | 0.433 | 0.411 | 0.433 | 0.418 | 0.431 | 0.407 | 0.490 | 0.449 |
| PJM | **0.064** | **0.155** | 0.100 | 0.198 | 0.098 | 0.203 | 0.079 | 0.177 | 0.120 | 0.246 | **0.066** | **0.158** | 0.102 | 0.199 | 0.101 | 0.201 | 0.079 | 0.176 | 0.112 | 0.224 |
| NP | **0.204** | **0.235** | 0.289 | 0.296 | 0.426 | 0.416 | 0.258 | 0.270 | 0.268 | 0.293 | **0.208** | **0.236** | 0.293 | 0.297 | 0.248 | 0.285 | 0.260 | 0.270 | 0.269 | 0.291 |

models channel relationships and can leverage historical exogenous information, also performs well. In contrast, PatchTST performs poorly because it cannot exploit historical exogenous variables.

**Robustness under Missing Values** In complex real-world scenarios, time series data often contain noises due to uncontrollable factors. To further evaluate the generalizability of GCGNet under such conditions, we simulate missing values in exogenous variables by randomly masking the original time series. Specifically, we adopt two masking strategies to assess GCGNet's adaptability: (1) **Zeros**: masked values are replaced with 0; and (2) **Random**: masked values are replaced with random values sampled from the normal distribution $\mathcal{N}(0, 1)$. We conduct experiments under three masking ratios: 10%, 30%, and 50%. Table 4 reports the average results under three masking ratios, while the full results are provided in Table 10 of Appendix B.4. The experimental results show that GCGNet continues to perform excellently, demonstrating that the design of the generative network in GCGNet not only ensures accurate forecasting but also provides robustness in real-world scenarios where the data may contain noises.

**Effect of the Generative Network** To assess the contribution of the generative network in GCGNet, we compare the original model with a variant in which all VAE modules are replaced by MLPs (denoted as w/o VAE in the table). Experiments follow the settings in Section 4.2. Table 5 reports average results across multiple datasets under partially missing exogenous variables. Across all datasets, GCGNet performs better than the version without the VAE. This improvement shows that the generative network helps the model handle missing or noisy inputs,

Table 5: Average results of GCGNet and the version without VAE under the setting where exogenous variables are partial missing. The best results are highlighted in **bold**.

| Mask Type | Zero | | | | Random | | | |
|---|---|---|---|---|---|---|---|---|
| Models | GCGNet | | w/o VAE | | GCGNet | | w/o VAE | |
| Colbun | **0.076** | **0.093** | 0.094 | 0.140 | **0.070** | **0.096** | 0.081 | 0.123 |
| Energy | **0.070** | **0.203** | 0.142 | 0.293 | **0.081** | **0.217** | 0.142 | 0.291 |
| DE | **0.292** | **0.336** | 0.453 | 0.430 | **0.344** | **0.366** | 0.505 | 0.460 |
| PJM | **0.064** | **0.155** | 0.108 | 0.208 | **0.066** | **0.158** | 0.094 | 0.193 |
| NP | **0.204** | **0.235** | 0.301 | 0.310 | **0.208** | **0.236** | 0.272 | 0.283 |

recover useful structure from incomplete data, and provide more stable forecasts. These results also

suggest that the VAE captures hidden correlations that simple architectures cannot, which leads to better overall forecasting accuracy.

**The Advantages of Joint Modeling** To validate the motivation of GCGNet, we visualize its predictions on the NP dataset, where the target variable is electricity price (Figure 4c) and the exogenous variables are wind power (Figure 4a) and grid load (Figure 4b). PatchTST and CrossLinear serve as comparison models. As shown in Figure 4d, as the historical endogenous data exhibit clear periodic patterns, PatchTST, unable to consider historical exogenous variables, mainly replicates historical endogenous data patterns. Meanwhile, as shown in Figure 4e, CrossLinear, which models channel correlations first and then temporal correlations, causes the two types of correlations to interfere with each other, resulting in neither type of information being well captured. In contrast, GCGNet generates predictions (Figure 4f) that closely match the ground truth. This demonstrates that GCGNet effectively uses both endogenous and historical exogenous variables by jointly modeling temporal and channel correlations. When future exogenous variables are included (Figure 4i), GCGNet's performance improves further. While in Figure 4g, an MLP fusion module (Appendix A.3) is applied to PatchTST, enabling it to incorporate future exogenous variables in a two-step modeling strategy. The first step models temporal correlations, and the second step models channel correlations. We observe that this approach provides only limited improvement, as its predictions closely follow the shape of the future grid load, showing that correlations learned in the second step can override those from the first step. A similar behavior can also be observed for CrossLinear in Figure 4h. Additional cases are provided in Appendix B.1. These results demonstrate the effectiveness of GCGNet's joint modeling strategy in forecasting with exogenous variables.

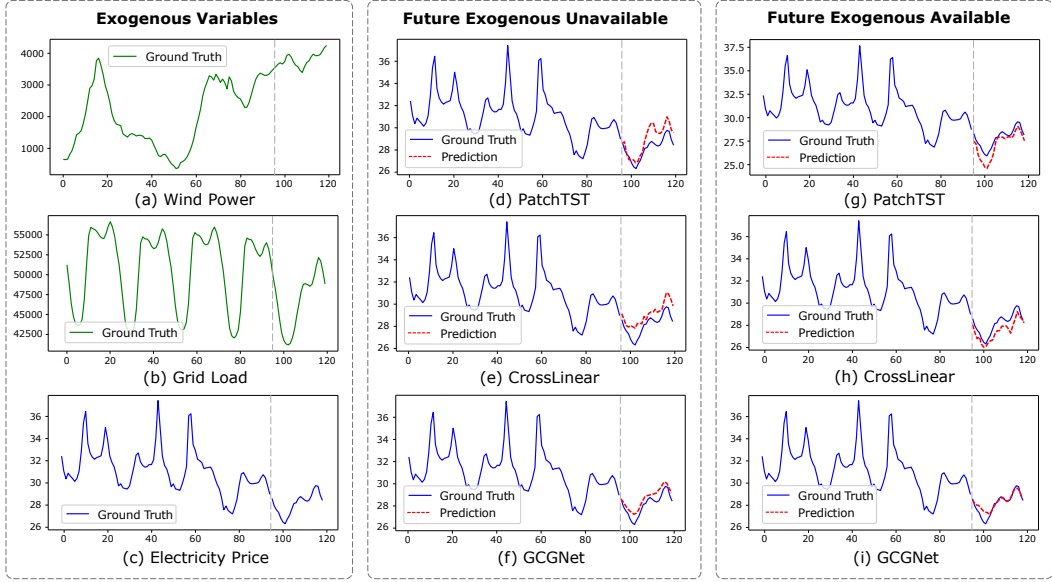

Figure 4: Prediction results on the NP dataset. (a–b) Exogenous variables: wind power and grid load. (c) Endogenous variables: Electricity Price. (d–f) PatchTST, CrossLinear and GCGNet predictions when future exogenous variables are unavailable. (g–i) PatchTST, CrossLinear and GCGNet predictions when future exogenous variables are available.

## 5 CONCLUSION

In this paper, we introduce **GCGNet**, a graph-consistent generative network designed for time series forecasting with exogenous variables. To summarize, our method first generates coarse predictions, then enforces structural alignment to capture joint temporal and channel correlations robustly, and finally refines the predictions to prevent degeneration and improve accuracy. These designs collectively enable GCGNet to capture complex joint temporal and channel correlations while maintaining robustness in real-world scenarios. Extensive experiments on diverse real-world datasets demonstrate that GCGNet achieves state-of-the-art forecasting performance.

## ACKNOWLEDGEMENTS

This work was partially supported by the National Natural Science Foundation of China (No. 62372179, No. 62472174) and the Fundamental Research Funds for the Central Universities. Bin Yang is the corresponding author of the work.

## ETHICS STATEMENT

This study relies solely on publicly available datasets that contain no personally identifiable information. The proposed framework is designed for socially beneficial applications in time series forecasting. No human subjects were involved in this research.

## REPRODUCIBILITY STATEMENT

The promise that all experimental results can be reproduced. We have released our model code in an anonymous repository: https://github.com/decisionintelligence/GCGNet.

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

# A  EXPERIMENTAL DETAILS

## A.1  DATASETS

Table 6: Statistics of datasets. Ex. and En. are abbreviations for the Exogenous variables and Endogenous variables, respectively.

| Dataset | #Num | Sampling Frequency | Lengths | Split | Ex. Descriptions | En. Descriptions |
|---|---|---|---|---|---|---|
| NP | 2 | 1 Hour | 52,416 | 7:1:2 | Grid Load, Wind Power | Nord Pool Electricity Price |
| PJM | 2 | 1 Hour | 52,416 | 7:1:2 | System Load, SyZonal COMED load | Pennsylvania-New Jersey-MarylandElectricity Price |
| BE | 2 | 1 Hour | 52,416 | 7:1:2 | Generation, System Load | Belgium's Electricity Price |
| FR | 2 | 1 Hour | 52,416 | 7:1:2 | Generation, System Load | France's Electricity Price |
| DE | 2 | 1 Hour | 52,416 | 7:1:2 | Wind power, Amprion zonal load | German's Electricity Price |
| Energy | 5 | 1 Hour | 13,064 | 7:1:2 | Battery, Geothermal, Hydroelectric, Solar, Wind | Thermoelectric |
| Colbún | 2 | 1 Day | 2,958 | 7:1:2 | Precipitation, Tributary Inflow | Water Level |
| Rapel | 2 | 1 Day | 3,366 | 7:1:2 | Precipitation, Tributary Inflow | Water Level |
| Sdwpfh | 6 | 1 Hour | 14,641 | 7:1:2 | Climate Feature | Active Power |
| Sdwpfm | 6 | 30 Minutes | 29,281 | 7:1:2 | Climate Feature | Active Power |

We conduct experiments on 12 real-world datasets with exogenous variables, where future exogenous variables are either approximately known or can be obtained with high accuracy. These include 5 EPF (Lago et al., 2021; Wang et al., 2024) datasets (NP, PJM, BE, FR, DE) and 7 datasets collected by DAG (Qiu et al., 2025e) (Energy, Colbún, Rapel, Sdwpfh1, Sdwpfh2, Sdwpfm1, Sdwpfm2). **NP** records Nord Pool market with hourly electricity price as the endogenous variable and grid load and wind power forecast as exogenous features. **PJM** records the Pennsylvania–New Jersey–Maryland Interconnection market with zonal electricity price in the COMED area as the endogenous variable, and system load and COMED load forecasts as exogenous variables. **BE** records Belgium's electricity market with hourly price as the endogenous variable and load forecast in Belgium plus generation forecast from France as exogenous variables. **FR** records the French electricity market with price as the endogenous variable and generation and load forecasts as exogenous variables. **DE** records the German electricity market with hourly price as the endogenous variable and zonal load forecast in Amprion, along with wind and solar generation forecasts as exogenous variables. **Energy** provides hourly power generation data from Chile, with thermoelectric generation as the endogenous variable and battery storage, wind, hydro, and solar as exogenous variables. **Colbún and Rapel** are daily-resolution hydropower datasets with water level as the endogenous variable and precipitation and tributary inflow as exogenous variables. **Sdwpfh1, Sdwpfh2, Sdwpfm1, Sdwpfm2** are wind power datasets from Longyuan wind farm, with active power output (Patv) as the endogenous variable and external weather data from ERA5 (Hersbach et al., 2020) as exogenous variables, including temperature, surface pressure, relative humidity, wind speed, wind direction, and total precipitation. It is also worth noting that modern deep learning research typically evaluates models on a diverse collection of datasets to demonstrate generalization across different application domains (Ma et al., 2024a;b; 2025e; Yang et al., 2023; Wu et al., 2024). This evaluation paradigm is widely adopted not only in natural language processing and computer vision (Ma et al., 2025d;c;b), but also in time series analysis under various problem settings, including forecasting (Qiu et al., 2025c; Shang et al., 2024; Shang & Chen, 2024; Wang et al., 2026b;c; 2025c;b), anomaly detection (Wu et al., 2025c), imputation (Wang et al., 2025a), modeling irregular (Liu et al., 2026b; Yu et al., 2025b), long-term modeling (Liu et al., 2026b; Yu et al., 2025b), and general modeling frameworks (Wu et al., 2026; Shang et al., 2026). Following this evaluation paradigm, the datasets considered in this work cover multiple real-world domains with heterogeneous data characteristics, thereby enabling a comprehensive evaluation of model performance under diverse conditions.

## A.2  IMPLEMENTATION DETAILS

The "*Drop Last*" issue is reported by several researchers (Qiu et al., 2024; 2025b; Li et al., 2025). That is, in some previous works evaluating the model on the test set with drop-last=True setting may cause additional errors related to test batch size. In our experiment, to ensure a fair comparison, we set drop-last to False for all baselines to avoid this issue. All experiments are conducted using PyTorch (Paszke et al., 2019) in Python 3.8 and execute on an NVIDIA Tesla-A800 GPU. We do not use the "*Drop Last*" operation during testing. To ensure reproducibility and facilitate experimentation, datasets and code are available at: https://github.com/decisionintelligence/GCGNet.

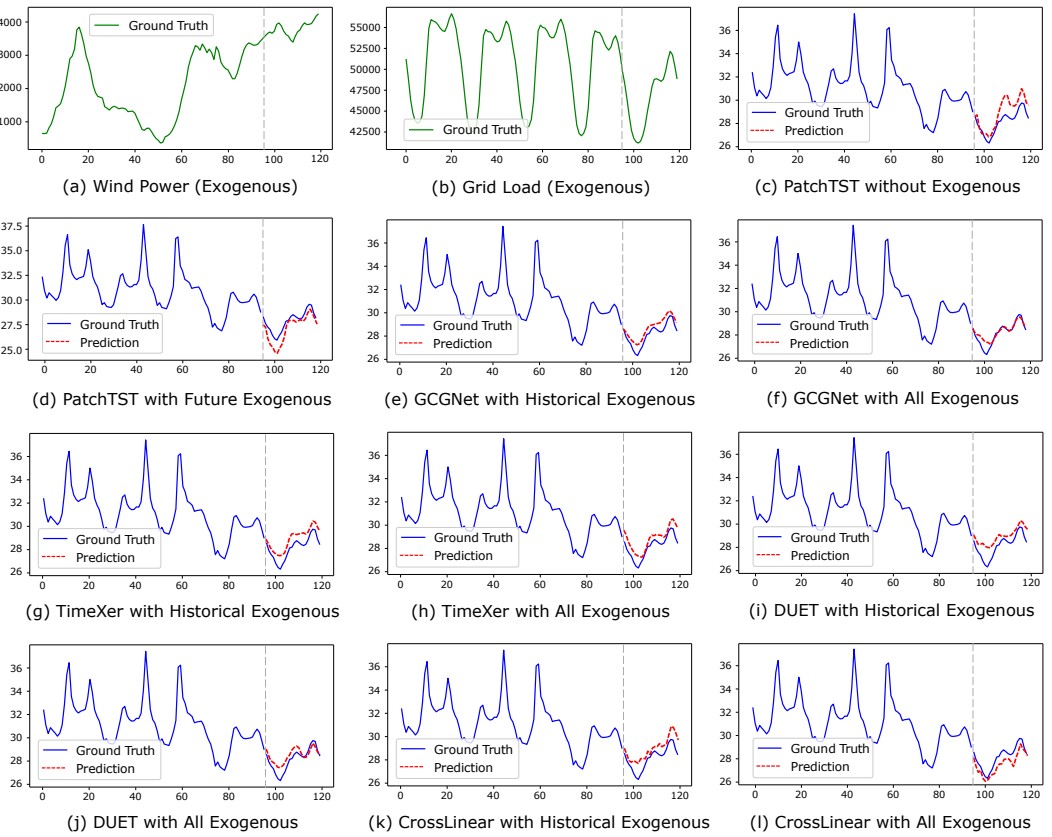

Figure 5: Visualization of prediction results on NP dataset.

### A.3 MLP FUSION MODULE

We employ an MLP fusion module, which is efficient (Huang et al., 2025a), to enable traditional forecasting methods to leverage future exogenous variables for a fairer comparison. Given historical endogenous variables $X^{\text{endo}} \in \mathbb{R}^{N \times T}$ and historical exogenous variables $X^{\text{exo}} \in \mathbb{R}^{D \times T}$, the backbone model first generates a latent representation:

$$z = \theta_{\text{Model}}(X^{\text{endo}}, X^{\text{exo}}), \tag{14}$$

where $z \in \mathbb{R}^{N \times F}$, $\theta_{\text{Model}}$ denotes the parameters of the forecasting model. Subsequently, the latent representation $z$ is concatenated with the future exogenous variables $Y^{\text{exo}} \in \mathbb{R}^{D \times F}$ and passed through an MLP to produce the final prediction:

$$\hat{Y}^{\text{endo}} = \theta_{\text{MLP}}\big(\text{Concat}(z, Y^{\text{exo}})\big), \tag{15}$$

where $\theta_{\text{MLP}}$ are the parameters of the fusion module.

## B EXPERIMENTAL RESULTS

### B.1 VISUALIZATION OF RESULTS

We visualize more cases of predictions on the NP dataset in Figure 5, where the target variable is electricity price and the exogenous variables are wind power (Figure 5a) and grid load (Figure 5b). As shown in Figure 5, GCGNet consistently outperforms all baselines, both when future exogenous variables are available and when they are not, indicating that it effectively captures joint temporal and channel correlations. As shown in Figure 5c, PatchTST, which models channels independently, cannot incorporate exogenous variables and therefore mainly reproduces the patterns of historical endogenous variables. As shown in Figure 5d, when an MLP fusion module (Appendix A.3) is

added to PatchTST, it can include future exogenous variables through a two-step modeling strategy. However, this only brings limited improvement, as its predictions largely follow the shape of the future grid load, showing that the modeling in the two steps interferes with each other. As shown in Figure 5g and Figure 5i, in the setting with only historical exogenous variables, TimeXer and DUET can leverage historical exogenous variables and thus go beyond simply repeating past endogenous patterns. In contrast, Figure 5k shows that CrossLinear first models channel dependencies and then temporal correlations. Since its first step uses a coarse-grained 1D convolution for channel modeling, it may distort the periodic information of historical endogenous variables, leading to predictions that deviate from the true patterns. As shown in Figure 5h, Figure 5j, and Figure 5i, when future exogenous variables are introduced, DUET, CrossLinear, and TimeXer all achieve improvements, but these gains remain limited, further confirming the effectiveness of GCGNet.

## B.2 FORECASTING WITHOUT FUTURE EXOGENOUS VARIABLES

Considering that future exogenous variables may not always be available in practice, we conduct experiments using only historical endogenous and exogenous variables to assess the robustness and applicability of GCGNet. The results in Table 7 indicate that GCGNet maintains strong performance even without future exogenous variables, highlighting its robustness and generalizability.

## B.3 CONTROLLED HYPERPARAMETER EXPERIMENT

We also conduct all experiments under controlled hyperparameter settings, where each model uses a fixed configuration across forecasting horizons. The results under the setting where future exogenous variables are available are reported in Table 8, while the results without future exogenous variables are presented in Table 9. It can be observed that GCGNet achieves the best performance across both settings.

## B.4 FORECASTING WITH MISSING EXOGENOUS VARIABLES

In complex real-world scenarios, time series data are often incomplete due to uncontrollable factors such as sensor failures, making robust forecasting models essential. To further evaluate the generalizability of GCGNet under such conditions, we simulate missing values in exogenous variables by randomly masking the original time series. Specifically, we adopt two masking strategies to assess GCGNet's adaptability: (1) **Zeros**: masked values are replaced with 0; and (2) **Random**: masked values are replaced with random values sampled from the normal distribution $\mathcal{N}(0, 1)$. We conduct experiments under three masking ratios: 10%, 30%, and 50%. We select TimeXer, CrossLinear, DUET, and PatchTST as baselines for comparison. Specifically, TimeXer and CrossLinear are representative methods designed to model interactions between endogenous and exogenous variables, DUET is a channel-dependent model, and PatchTST is a channel-independent model. The experimental results are reported in Table 10, where it can be observed that GCGNet consistently achieves the best performance across different masking strategies and ratios.

## B.5 PARAMETER SENSITIVITY

We also study the parameter sensitivity of the GCGNet, where all datasets use a lookback length of 720 and a forecasting horizon of 360 steps. We make the following observations: 1) Figure 6a

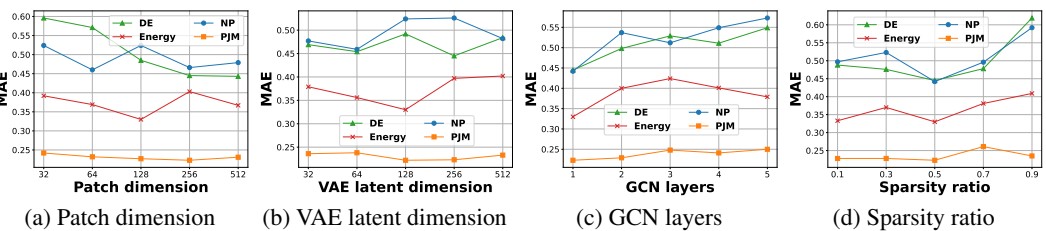

(a) Patch dimension  (b) VAE latent dimension  (c) GCN layers  (d) Sparsity ratio

Figure 6: Parameter sensitivity studies of main hyper-parameters in GCGNet.

Table 7: Results under the setting where future exogenous variables are not available. The inputs are ($X^{\text{endo}}$ and $X^{\text{exo}}$). The best results are **Red**, and the second-best results are Blue. Avg represents the average results across the two forecasting horizons.

| | Models | GCGNet | | TimeXer | | TFT | | TiDE | | DUET | | CrossLinear | | Amplifier | | TimeKAN | | xPatch | | PatchTST | |
|---|---|---|---|---|---|---|---|---|---|---|---|---|---|---|---|---|---|---|---|---|---|
| | Metrics | mse | mae | mse | mae | mse | mae | mse | mae | mse | mae | mse | mae | mse | mae | mse | mae | mse | mae | mse | mae |
| NP | 24 | **0.234** | **0.259** | 0.270 | 0.292 | 0.330 | 0.321 | 0.297 | 0.309 | 0.262 | 0.271 | 0.246 | 0.282 | 0.302 | 0.321 | 0.336 | 0.344 | 0.260 | 0.277 | 0.283 | 0.296 |
| | 360 | 0.617 | 0.495 | **0.609** | **0.474** | 0.964 | 0.655 | 0.638 | 0.523 | 0.626 | 0.494 | 0.655 | 0.505 | 0.738 | 0.574 | 0.632 | 0.527 | 0.717 | 0.528 | 0.630 | 0.505 |
| | Avg | **0.425** | **0.377** | 0.440 | 0.383 | 0.647 | 0.488 | 0.467 | 0.416 | 0.444 | 0.383 | 0.451 | 0.394 | 0.520 | 0.448 | 0.484 | 0.435 | 0.488 | 0.402 | 0.457 | 0.401 |
| PJM | 24 | **0.073** | **0.168** | 0.096 | 0.191 | 0.129 | 0.231 | 0.105 | 0.213 | 0.082 | 0.178 | 0.096 | 0.198 | 0.093 | 0.195 | 0.130 | 0.236 | 0.085 | 0.178 | 0.106 | 0.212 |
| | 360 | 0.192 | **0.266** | **0.186** | 0.267 | 0.270 | 0.310 | 0.210 | 0.292 | 0.198 | 0.273 | 0.197 | 0.283 | 0.210 | 0.294 | 0.219 | 0.304 | 0.193 | 0.267 | 0.189 | 0.273 |
| | Avg | **0.133** | **0.217** | 0.141 | 0.229 | 0.200 | 0.270 | 0.158 | 0.253 | 0.140 | 0.226 | 0.147 | 0.241 | 0.152 | 0.245 | 0.175 | 0.270 | 0.139 | 0.223 | 0.148 | 0.243 |
| BE | 24 | **0.363** | **0.251** | 0.392 | 0.253 | 0.400 | 0.267 | 0.492 | 0.322 | 0.376 | 0.253 | 0.389 | 0.253 | 0.417 | 0.283 | 0.424 | 0.280 | 0.404 | 0.258 | 0.405 | 0.264 |
| | 360 | 0.553 | 0.351 | 0.563 | 0.349 | 0.727 | 0.442 | 0.602 | 0.375 | 0.571 | 0.357 | 0.564 | **0.346** | 0.588 | 0.367 | **0.552** | 0.350 | 0.562 | 0.346 | 0.564 | 0.354 |
| | Avg | **0.458** | 0.301 | 0.477 | 0.301 | 0.563 | 0.354 | 0.547 | 0.348 | 0.473 | 0.305 | 0.477 | **0.300** | 0.502 | 0.325 | 0.488 | 0.315 | 0.483 | 0.302 | 0.485 | 0.309 |
| FR | 24 | **0.338** | **0.194** | 0.366 | 0.195 | 0.445 | 0.231 | 0.415 | 0.254 | 0.359 | 0.208 | 0.397 | 0.204 | 0.429 | 0.250 | 0.448 | 0.250 | 0.402 | 0.208 | 0.397 | 0.223 |
| | 360 | **0.519** | **0.296** | 0.542 | 0.300 | 0.624 | 0.345 | 0.574 | 0.325 | 0.577 | 0.317 | 0.555 | 0.310 | 0.559 | 0.322 | 0.534 | 0.303 | 0.553 | 0.306 | 0.542 | 0.306 |
| | Avg | **0.428** | **0.245** | 0.454 | 0.247 | 0.535 | 0.288 | 0.494 | 0.290 | 0.468 | 0.262 | 0.476 | 0.257 | 0.494 | 0.286 | 0.491 | 0.276 | 0.477 | 0.257 | 0.470 | 0.264 |
| DE | 24 | **0.415** | **0.400** | 0.501 | 0.445 | 0.576 | 0.452 | 0.465 | 0.433 | 0.424 | 0.403 | 0.438 | 0.423 | 0.493 | 0.455 | 0.504 | 0.462 | 0.455 | 0.425 | 0.503 | 0.450 |
| | 360 | 0.808 | 0.594 | 0.818 | **0.568** | **0.792** | 0.578 | 0.823 | 0.604 | 0.895 | 0.623 | 0.832 | 0.592 | 0.930 | 0.642 | 0.834 | 0.590 | 0.810 | 0.575 | 0.889 | 0.604 |
| | Avg | **0.611** | **0.497** | 0.659 | 0.507 | 0.684 | 0.515 | 0.644 | 0.519 | 0.660 | 0.513 | 0.635 | 0.508 | 0.712 | 0.548 | 0.669 | 0.526 | 0.633 | 0.500 | 0.696 | 0.527 |
| Energy | 24 | **0.102** | **0.246** | 0.138 | 0.293 | 0.352 | 0.463 | 0.117 | 0.265 | 0.108 | 0.254 | 0.102 | 0.246 | 0.108 | 0.254 | 0.109 | 0.254 | 0.107 | 0.252 | 0.108 | 0.254 |
| | 360 | 0.197 | 0.352 | 0.206 | 0.360 | 0.399 | 0.497 | 0.207 | 0.358 | 0.211 | 0.362 | 0.299 | 0.425 | 0.251 | 0.400 | **0.186** | **0.338** | 0.232 | 0.377 | 0.298 | 0.428 |
| | Avg | 0.150 | 0.299 | 0.172 | 0.326 | 0.376 | 0.480 | 0.162 | 0.311 | 0.160 | 0.308 | 0.201 | 0.336 | 0.180 | 0.327 | **0.147** | **0.296** | 0.170 | 0.314 | 0.203 | 0.341 |
| Sdwpfm1 | 24 | **0.536** | **0.493** | 0.558 | 0.533 | 0.596 | 0.531 | 0.561 | 0.540 | 0.550 | 0.495 | 0.549 | 0.553 | 0.579 | 0.557 | 0.598 | 0.579 | 0.564 | 0.509 | 0.557 | 0.557 |
| | 360 | 0.864 | 0.712 | **0.845** | **0.684** | 1.372 | 0.925 | 0.864 | 0.725 | 0.898 | 0.672 | 1.070 | 0.834 | 1.106 | 0.814 | 0.853 | 0.764 | 0.910 | 0.672 | 0.910 | 0.745 |
| | Avg | **0.700** | 0.602 | 0.701 | 0.609 | 0.984 | 0.728 | 0.713 | 0.633 | 0.724 | **0.583** | 0.809 | 0.694 | 0.843 | 0.685 | 0.725 | 0.672 | 0.737 | 0.590 | 0.733 | 0.651 |
| Sdwpfm2 | 24 | 0.614 | 0.548 | 0.627 | 0.570 | 0.749 | 0.657 | 0.636 | 0.582 | **0.612** | **0.537** | 0.635 | 0.586 | 0.669 | 0.600 | 0.684 | 0.621 | 0.641 | 0.539 | 0.648 | 0.610 |
| | 360 | 1.036 | **0.717** | 0.978 | 0.736 | 1.340 | 0.877 | 0.995 | 0.783 | 1.028 | 0.745 | **0.965** | 0.788 | 1.215 | 0.863 | 0.987 | 0.780 | 1.058 | 0.726 | 1.020 | 0.819 |
| | Avg | 0.825 | 0.633 | 0.803 | 0.653 | 1.044 | 0.767 | 0.816 | 0.682 | 0.820 | 0.641 | **0.800** | 0.687 | 0.942 | 0.732 | 0.836 | 0.701 | 0.850 | **0.632** | 0.834 | 0.714 |
| Sdwpfh1 | 24 | 0.668 | 0.583 | **0.651** | 0.587 | 0.754 | 0.670 | 0.719 | 0.641 | 0.669 | **0.572** | 0.687 | 0.652 | 1.111 | 0.816 | 0.755 | 0.679 | 0.701 | 0.579 | 0.669 | 0.629 |
| | 360 | 0.846 | **0.674** | 0.841 | 0.700 | **0.831** | 0.727 | 0.897 | 0.757 | 0.890 | 0.702 | 0.849 | 0.752 | 1.089 | 0.823 | 0.853 | 0.762 | 0.882 | 0.680 | 0.939 | 0.804 |
| | Avg | 0.757 | **0.628** | **0.746** | 0.643 | 0.793 | 0.698 | 0.808 | 0.699 | 0.780 | 0.637 | 0.768 | 0.702 | 1.100 | 0.820 | 0.804 | 0.720 | 0.791 | 0.630 | 0.804 | 0.717 |
| Sdwpfh2 | 24 | 0.797 | 0.644 | 0.820 | 0.677 | 0.811 | 0.714 | 0.783 | 0.674 | 0.805 | **0.629** | 0.810 | 0.718 | 0.834 | 0.691 | 0.892 | 0.740 | 0.829 | 0.632 | **0.769** | 0.680 |
| | 360 | 0.968 | **0.739** | **0.962** | 0.761 | 1.042 | 0.779 | 1.056 | 0.829 | 1.036 | 0.754 | 1.102 | 0.830 | 1.126 | 0.845 | 0.990 | 0.825 | 1.053 | 0.742 | 1.280 | 0.925 |
| | Avg | **0.882** | 0.691 | 0.891 | 0.719 | 0.926 | 0.746 | 0.919 | 0.751 | 0.920 | 0.691 | 0.956 | 0.774 | 0.980 | 0.768 | 0.941 | 0.782 | 0.941 | **0.687** | 1.025 | 0.802 |
| Colbun | 10 | **0.064** | **0.081** | 0.084 | 0.146 | 0.449 | 0.279 | 0.095 | 0.140 | 0.073 | 0.106 | 0.068 | 0.101 | 0.077 | 0.120 | 0.091 | 0.129 | 0.071 | 0.092 | 0.083 | 0.125 |
| | 30 | **0.168** | **0.246** | 0.181 | 0.293 | 0.663 | 0.494 | 0.282 | 0.341 | 0.221 | 0.290 | 0.190 | 0.289 | 0.227 | 0.301 | 0.311 | 0.377 | 0.405 | 0.379 | 0.217 | 0.316 |
| | Avg | **0.116** | **0.164** | 0.132 | 0.219 | 0.556 | 0.386 | 0.188 | 0.240 | 0.147 | 0.198 | 0.129 | 0.195 | 0.152 | 0.210 | 0.201 | 0.253 | 0.238 | 0.235 | 0.150 | 0.221 |
| Rapel | 10 | **0.191** | **0.204** | 0.301 | 0.308 | 0.238 | 0.274 | 0.238 | 0.270 | 0.221 | 0.225 | 0.201 | 0.224 | 0.240 | 0.259 | 0.226 | 0.238 | 0.421 | 0.421 | 0.215 | 0.232 |
| | 30 | 0.325 | **0.343** | 0.387 | 0.416 | 0.377 | 0.394 | 0.409 | 0.444 | 0.387 | 0.386 | **0.278** | 0.374 | 0.433 | 0.436 | 0.458 | 0.437 | 0.307 | 0.343 | 0.436 | 0.436 |
| | Avg | 0.258 | **0.273** | 0.344 | 0.362 | 0.308 | 0.334 | 0.323 | 0.357 | 0.304 | 0.306 | **0.240** | 0.299 | 0.337 | 0.348 | 0.342 | 0.337 | 0.364 | 0.382 | 0.325 | 0.334 |
| 1st Count | | **19** | **23** | 6 | 2 | 2 | 0 | 0 | 0 | 1 | 5 | 4 | 2 | 0 | 0 | 3 | 2 | 0 | 2 | 1 | 0 |

and Figure 6b illustrate the performance of GCGNet under different settings of the patch dimension and the VAE latent dimension. The results indicate that larger dimensions do not always improve forecasting accuracy because excessively high dimensions can introduce redundancy, increase overfitting risk, and make optimization more difficult. Instead, dimensions in the range of 64 to 256 generally achieve more favorable performance. 2) Figure 6c shows that a shallower GCN architecture with one or two layers performs better. This suggests that the node relationships in the task are relatively direct, and deeper networks may suffer from the over smoothing problem, thereby degrading the forecasting results. 3) Figure 6d shows that the model achieves better performance when the sparsity ratio is set to 50%. This indicates that a 50% sparsity strikes a good balance, effectively removing most of the uninformative weights while retaining the critical connections necessary for accurate forecasting.

## B.6 Increasing Look-back Length

In general, increasing the look-back window provides the model with more historical information, which can improve forecasting accuracy. However, a longer look-back window also introduces more noises, particularly in forecasting with exogenous variables, which introduces additional noises. To assess the effectiveness of GCGNet, we configure different look-back windows on the DE, PJM, Energy, and NP datasets. We also visualize prediction results for look-back windows of 96, 168, 336, 512, and 720, with a forecasting horizon of 24. As shown in Figure 7, GCGNet consistently

Table 8: Results under the setting where future exogenous variables are available and each model uses a fixed configuration across forecasting horizons. The inputs are ($X^{\text{endo}}$, $Y^{\text{endo}}$ and $X^{\text{exo}}$). The best results are **Red**, and the second-best restuls are Blue. Avg represents the average results across the two forecasting horizons.

| Models Metrics | | GCGNet mse | mae | TimeXer mse | mae | TFT mse | mae | TiDE mse | mae | DUET mse | mae | CrossLinear mse | mae | Amplifier mse | mae | TimeKAN mse | mae | xPatch mse | mae | PatchTST mse | mae |
|---|---|---|---|---|---|---|---|---|---|---|---|---|---|---|---|---|---|---|---|---|---|
| NP | 24 | **0.208** | **0.237** | 0.236 | 0.266 | 0.219 | 0.249 | 0.284 | 0.301 | 0.246 | 0.287 | 0.210 | 0.266 | 0.252 | 0.303 | 0.273 | 0.310 | 0.234 | 0.278 | 0.249 | 0.294 |
| | 360 | 0.531 | **0.459** | 0.600 | 0.475 | 0.539 | 0.501 | 0.601 | 0.498 | 0.576 | 0.528 | 0.531 | 0.508 | 0.587 | 0.534 | 0.538 | 0.529 | **0.523** | 0.461 | 0.530 | 0.498 |
| | Avg | **0.370** | **0.348** | 0.418 | 0.371 | 0.379 | 0.375 | 0.443 | 0.400 | 0.411 | 0.408 | 0.371 | 0.387 | 0.420 | 0.418 | 0.405 | 0.419 | 0.378 | 0.370 | 0.390 | 0.396 |
| PJM | 24 | **0.060** | **0.150** | 0.075 | 0.166 | 0.095 | 0.195 | 0.106 | 0.214 | 0.072 | 0.166 | 0.088 | 0.191 | 0.096 | 0.208 | 0.115 | 0.244 | 0.077 | 0.168 | 0.116 | 0.239 |
| | 360 | **0.129** | 0.223 | 0.140 | 0.231 | 0.133 | 0.219 | 0.177 | 0.279 | 0.131 | 0.228 | 0.135 | 0.254 | 0.177 | 0.285 | 0.162 | 0.281 | 0.131 | 0.220 | 0.149 | 0.279 |
| | Avg | **0.095** | **0.187** | 0.108 | 0.198 | 0.114 | 0.207 | 0.142 | 0.246 | 0.102 | 0.197 | 0.112 | 0.223 | 0.137 | 0.246 | 0.139 | 0.262 | 0.104 | 0.194 | 0.133 | 0.259 |
| BE | 24 | **0.350** | **0.248** | 0.392 | 0.253 | 0.426 | 0.272 | 0.426 | 0.285 | 0.432 | 0.272 | 0.391 | 0.259 | 0.471 | 0.339 | 0.451 | 0.319 | 0.411 | 0.256 | 0.452 | 0.326 |
| | 360 | 0.511 | 0.340 | 0.512 | 0.327 | 0.482 | 0.310 | 0.571 | 0.364 | 0.597 | 0.436 | 0.568 | 0.416 | 0.646 | 0.487 | 0.645 | 0.495 | 0.579 | 0.410 | 0.702 | 0.538 |
| | Avg | **0.431** | 0.294 | 0.452 | **0.290** | 0.454 | 0.291 | 0.498 | 0.325 | 0.515 | 0.354 | 0.479 | 0.337 | 0.559 | 0.413 | 0.548 | 0.407 | 0.495 | 0.333 | 0.577 | 0.432 |
| FR | 24 | **0.347** | **0.188** | 0.366 | 0.208 | 0.543 | 0.253 | 0.418 | 0.255 | 0.384 | 0.251 | 0.390 | 0.226 | 0.459 | 0.348 | 0.454 | 0.296 | 0.424 | 0.235 | 0.518 | 0.368 |
| | 360 | 0.482 | 0.279 | 0.489 | 0.273 | 0.465 | 0.261 | 0.551 | 0.308 | 0.607 | 0.403 | 0.575 | 0.370 | 0.648 | 0.452 | 0.641 | 0.452 | 0.556 | 0.350 | 0.658 | 0.452 |
| | Avg | **0.415** | **0.234** | 0.427 | 0.241 | 0.504 | 0.257 | 0.484 | 0.281 | 0.496 | 0.327 | 0.483 | 0.298 | 0.554 | 0.408 | 0.547 | 0.374 | 0.490 | 0.293 | 0.588 | 0.410 |
| DE | 24 | **0.280** | **0.331** | 0.339 | 0.362 | 0.380 | 0.383 | 0.367 | 0.383 | 0.376 | 0.378 | 0.387 | 0.396 | 0.394 | 0.407 | 0.399 | 0.412 | 0.399 | 0.394 | 0.413 | 0.412 |
| | 360 | **0.523** | **0.447** | 0.610 | 0.474 | 0.599 | 0.509 | 0.630 | 0.511 | 0.589 | 0.482 | 0.583 | 0.507 | 0.551 | 0.474 | 0.547 | 0.479 | 0.550 | 0.472 | 0.589 | 0.498 |
| | Avg | **0.401** | **0.389** | 0.475 | 0.418 | 0.489 | 0.446 | 0.499 | 0.447 | 0.482 | 0.430 | 0.485 | 0.452 | 0.473 | 0.441 | 0.473 | 0.445 | 0.474 | 0.433 | 0.501 | 0.455 |
| Energy | 24 | **0.081** | **0.221** | 0.122 | 0.273 | 0.093 | 0.235 | 0.103 | 0.248 | 0.117 | 0.283 | 0.241 | 0.418 | 0.138 | 0.306 | 0.135 | 0.298 | 0.092 | 0.240 | 0.239 | 0.390 |
| | 360 | 0.182 | 0.334 | 0.204 | 0.357 | 0.167 | 0.331 | 0.202 | 0.355 | 0.288 | 0.452 | 0.237 | 0.385 | 0.328 | 0.472 | 0.302 | 0.464 | 0.204 | 0.367 | 0.214 | 0.363 |
| | Avg | 0.131 | **0.277** | 0.163 | 0.315 | 0.130 | 0.283 | 0.153 | 0.302 | 0.203 | 0.367 | 0.239 | 0.402 | 0.233 | 0.389 | 0.218 | 0.381 | 0.148 | 0.303 | 0.226 | 0.377 |
| Sdwpfm1 | 24 | 0.376 | **0.415** | 0.558 | 0.533 | 0.366 | 0.421 | 0.474 | 0.488 | 0.551 | 0.564 | **0.355** | 0.473 | 0.364 | 0.445 | 0.418 | 0.503 | 0.669 | 0.580 | 0.374 | 0.454 |
| | 360 | **0.472** | **0.499** | 0.845 | 0.684 | 0.597 | 0.528 | 0.492 | 0.526 | 0.646 | 0.577 | 0.497 | 0.532 | 0.510 | 0.535 | 0.476 | 0.566 | 0.592 | 0.556 | 0.497 | 0.550 |
| | Avg | **0.424** | **0.457** | 0.701 | 0.609 | 0.482 | 0.474 | 0.483 | 0.507 | 0.599 | 0.570 | 0.426 | 0.502 | 0.437 | 0.490 | 0.447 | 0.534 | 0.630 | 0.568 | 0.435 | 0.502 |
| Sdwpfm2 | 24 | 0.421 | **0.441** | 0.627 | 0.570 | 0.411 | 0.458 | 0.461 | 0.492 | 0.445 | 0.452 | 0.477 | 0.536 | **0.394** | 0.462 | 0.474 | 0.538 | 0.508 | 0.478 | 0.418 | 0.492 |
| | 360 | 0.529 | 0.531 | 0.978 | 0.736 | 0.541 | 0.519 | 0.511 | 0.540 | 0.584 | 0.528 | 0.589 | 0.611 | 0.587 | 0.563 | 0.520 | 0.589 | 0.523 | 0.513 | 0.602 | 0.603 |
| | Avg | **0.475** | **0.486** | 0.803 | 0.653 | 0.476 | 0.488 | 0.486 | 0.516 | 0.514 | 0.490 | 0.533 | 0.573 | 0.491 | 0.512 | 0.497 | 0.564 | 0.516 | 0.495 | 0.510 | 0.547 |
| Sdwpfh1 | 24 | 0.435 | 0.486 | 0.651 | 0.587 | **0.401** | **0.460** | 0.434 | 0.489 | 0.527 | 0.513 | 0.548 | 0.585 | 0.576 | 0.627 | 0.511 | 0.582 | 0.524 | 0.505 | 0.422 | 0.505 |
| | 360 | **0.465** | 0.514 | 0.841 | 0.700 | 0.557 | 0.523 | 0.472 | 0.527 | 0.551 | 0.519 | 0.566 | 0.601 | 0.497 | 0.569 | 0.643 | 0.694 | 0.472 | 0.504 | 0.513 | 0.548 |
| | Avg | **0.450** | **0.500** | 0.746 | 0.643 | 0.479 | 0.491 | 0.453 | 0.508 | 0.539 | 0.516 | 0.557 | 0.593 | 0.537 | 0.598 | 0.577 | 0.638 | 0.498 | 0.505 | 0.468 | 0.527 |
| Sdwpfh2 | 24 | 0.473 | **0.506** | 0.820 | 0.677 | 0.474 | 0.493 | 0.579 | 0.553 | 0.629 | 0.563 | 0.468 | 0.540 | 0.473 | 0.533 | 0.580 | 0.614 | 0.640 | 0.554 | **0.457** | 0.527 |
| | 360 | **0.566** | 0.565 | 0.962 | 0.761 | 0.657 | 0.549 | 0.619 | 0.614 | 0.665 | 0.569 | 0.608 | 0.609 | 0.569 | 0.628 | 0.713 | 0.729 | 0.579 | 0.547 | 0.641 | 0.632 |
| | Avg | **0.520** | **0.536** | 0.891 | 0.719 | 0.566 | 0.521 | 0.599 | 0.583 | 0.647 | 0.566 | 0.538 | 0.574 | 0.521 | 0.581 | 0.647 | 0.672 | 0.610 | 0.551 | 0.549 | 0.580 |
| Colbun | 10 | 0.065 | 0.108 | 0.113 | 0.172 | 0.092 | 0.135 | 0.089 | 0.131 | 0.089 | 0.134 | 0.071 | 0.102 | 0.071 | 0.121 | **0.061** | **0.101** | 0.090 | 0.104 | 0.062 | 0.122 |
| | 30 | **0.149** | **0.243** | 0.176 | 0.299 | 0.383 | 0.460 | 0.240 | 0.322 | 0.307 | 0.397 | 0.182 | 0.288 | 0.275 | 0.370 | 0.195 | 0.249 | 0.217 | 0.315 | 0.417 | 0.496 |
| | Avg | **0.107** | **0.175** | 0.145 | 0.235 | 0.238 | 0.297 | 0.164 | 0.227 | 0.198 | 0.266 | 0.126 | 0.195 | 0.173 | 0.246 | 0.128 | 0.175 | 0.153 | 0.210 | 0.239 | 0.309 |
| Rapel | 10 | 0.211 | 0.230 | 0.301 | 0.308 | 0.201 | 0.253 | 0.228 | 0.271 | 0.174 | 0.219 | 0.163 | **0.209** | 0.181 | 0.227 | 0.174 | 0.231 | **0.161** | 0.216 | 0.163 | 0.218 |
| | 30 | 0.401 | **0.384** | 0.387 | 0.416 | 0.409 | 0.414 | 0.411 | 0.432 | 0.365 | 0.432 | 0.340 | 0.417 | 0.333 | 0.416 | 0.325 | 0.390 | 0.505 | 0.546 | 0.374 | 0.445 |
| | Avg | 0.306 | **0.307** | 0.344 | 0.362 | 0.305 | 0.333 | 0.320 | 0.351 | 0.269 | 0.326 | 0.252 | 0.313 | 0.257 | 0.321 | 0.249 | 0.311 | 0.333 | 0.381 | 0.269 | 0.332 |
| 1st Count | | **22** | **22** | 0 | 1 | 5 | 8 | 1 | 0 | 0 | 0 | 1 | 1 | 1 | 0 | 3 | 1 | 2 | 3 | 1 | 0 |

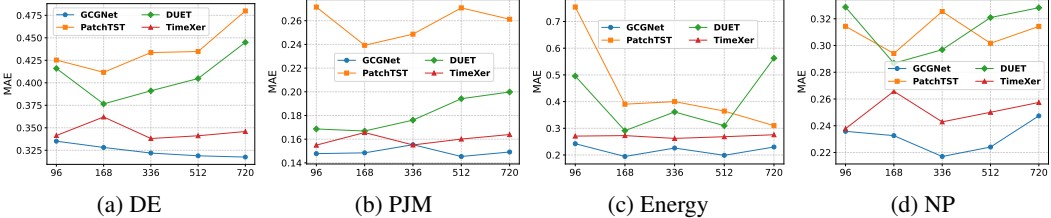

Figure 7: Forecasting performance under varying look-back windows across four datasets (DE, PJM, Energy, and NP).

(a) DE  (b) PJM  (c) Energy  (d) NP

outperforms the baselines across all four datasets. While other models exhibit large fluctuations in prediction performance as the look-back window increases, GCGNet maintains relatively stable metrics and even shows improvements as the look-back window increases, indicating its ability to model longer sequences and its robustness to the additional noises introduced by extended look-back windows.

Table 9: Results under the setting where future exogenous variables are not available and each model uses a fixed configuration across forecasting horizons. The inputs are ($X^{\text{endo}}$ and $X^{\text{exo}}$). The best results are **Red**, and the second-best restuls are Blue. Avg represents the average results across the two forecasting horizons.

| Models | | GCGNet | | TimeXer | | TFT | | TiDE | | DUET | | CrossLinear | | Amplifier | | TimeKAN | | xPatch | | PatchTST | |
|---|---|---|---|---|---|---|---|---|---|---|---|---|---|---|---|---|---|---|---|---|---|
| Metrics | | mse | mae | mse | mae | mse | mae | mse | mae | mse | mae | mse | mae | mse | mae | mse | mae | mse | mae | mse | mae |
| NP | 24 | 0.240 | 0.260 | 0.270 | 0.292 | 0.330 | 0.321 | 0.297 | 0.309 | 0.262 | 0.271 | 0.246 | 0.282 | 0.302 | 0.321 | 0.336 | 0.344 | 0.260 | 0.277 | 0.283 | 0.296 |
| | 360 | 0.620 | 0.502 | 0.609 | 0.474 | 0.964 | 0.655 | 0.638 | 0.523 | 0.626 | 0.494 | 0.655 | 0.505 | 0.738 | 0.574 | 0.632 | 0.527 | 0.717 | 0.528 | 0.630 | 0.505 |
| | Avg | 0.430 | 0.381 | 0.440 | 0.383 | 0.647 | 0.488 | 0.467 | 0.416 | 0.444 | 0.383 | 0.451 | 0.394 | 0.520 | 0.448 | 0.484 | 0.435 | 0.488 | 0.402 | 0.457 | 0.401 |
| PJM | 24 | 0.072 | 0.164 | 0.096 | 0.191 | 0.129 | 0.231 | 0.105 | 0.213 | 0.082 | 0.178 | 0.096 | 0.198 | 0.093 | 0.195 | 0.130 | 0.236 | 0.085 | 0.178 | 0.106 | 0.212 |
| | 360 | 0.191 | 0.269 | 0.186 | 0.267 | 0.270 | 0.310 | 0.210 | 0.292 | 0.198 | 0.273 | 0.197 | 0.283 | 0.210 | 0.294 | 0.219 | 0.304 | 0.193 | 0.267 | 0.189 | 0.273 |
| | Avg | 0.131 | 0.216 | 0.141 | 0.229 | 0.200 | 0.270 | 0.158 | 0.253 | 0.140 | 0.226 | 0.147 | 0.241 | 0.152 | 0.245 | 0.175 | 0.270 | 0.139 | 0.223 | 0.148 | 0.243 |
| BE | 24 | 0.365 | 0.259 | 0.392 | 0.253 | 0.400 | 0.267 | 0.492 | 0.322 | 0.376 | 0.253 | 0.389 | 0.253 | 0.417 | 0.283 | 0.424 | 0.280 | 0.404 | 0.258 | 0.405 | 0.264 |
| | 360 | 0.553 | 0.354 | 0.563 | 0.349 | 0.727 | 0.442 | 0.602 | 0.375 | 0.571 | 0.357 | 0.564 | 0.346 | 0.588 | 0.367 | 0.552 | 0.350 | 0.562 | 0.346 | 0.564 | 0.354 |
| | Avg | 0.459 | 0.306 | 0.477 | 0.301 | 0.563 | 0.354 | 0.547 | 0.348 | 0.473 | 0.305 | 0.477 | 0.300 | 0.502 | 0.325 | 0.488 | 0.315 | 0.483 | 0.302 | 0.485 | 0.309 |
| FR | 24 | 0.352 | 0.199 | 0.366 | 0.195 | 0.445 | 0.231 | 0.415 | 0.254 | 0.359 | 0.208 | 0.397 | 0.204 | 0.429 | 0.250 | 0.448 | 0.250 | 0.402 | 0.208 | 0.397 | 0.223 |
| | 360 | 0.519 | 0.293 | 0.542 | 0.300 | 0.624 | 0.345 | 0.574 | 0.325 | 0.577 | 0.317 | 0.555 | 0.310 | 0.559 | 0.322 | 0.534 | 0.303 | 0.553 | 0.306 | 0.542 | 0.306 |
| | Avg | 0.435 | 0.246 | 0.454 | 0.247 | 0.535 | 0.288 | 0.494 | 0.290 | 0.468 | 0.262 | 0.476 | 0.257 | 0.494 | 0.286 | 0.491 | 0.276 | 0.477 | 0.257 | 0.470 | 0.264 |
| DE | 24 | 0.413 | 0.402 | 0.501 | 0.445 | 0.576 | 0.452 | 0.465 | 0.433 | 0.424 | 0.403 | 0.438 | 0.423 | 0.493 | 0.455 | 0.504 | 0.462 | 0.455 | 0.425 | 0.503 | 0.450 |
| | 360 | 0.816 | 0.600 | 0.818 | 0.568 | 0.792 | 0.578 | 0.823 | 0.604 | 0.895 | 0.623 | 0.832 | 0.592 | 0.930 | 0.642 | 0.834 | 0.590 | 0.810 | 0.575 | 0.889 | 0.604 |
| | Avg | 0.614 | 0.501 | 0.659 | 0.507 | 0.684 | 0.515 | 0.644 | 0.519 | 0.660 | 0.513 | 0.635 | 0.508 | 0.712 | 0.548 | 0.669 | 0.526 | 0.633 | 0.500 | 0.696 | 0.527 |
| Energy | 24 | 0.112 | 0.259 | 0.138 | 0.293 | 0.352 | 0.463 | 0.117 | 0.265 | 0.108 | 0.254 | 0.102 | 0.246 | 0.108 | 0.254 | 0.109 | 0.254 | 0.107 | 0.252 | 0.108 | 0.254 |
| | 360 | 0.195 | 0.351 | 0.206 | 0.360 | 0.399 | 0.497 | 0.207 | 0.358 | 0.211 | 0.362 | 0.299 | 0.425 | 0.251 | 0.400 | 0.186 | 0.338 | 0.232 | 0.377 | 0.298 | 0.428 |
| | Avg | 0.154 | 0.305 | 0.172 | 0.326 | 0.376 | 0.480 | 0.162 | 0.311 | 0.160 | 0.308 | 0.201 | 0.336 | 0.180 | 0.327 | 0.147 | 0.296 | 0.170 | 0.314 | 0.203 | 0.341 |
| Sdwpfm1 | 24 | 0.537 | 0.509 | 0.558 | 0.533 | 0.596 | 0.531 | 0.561 | 0.540 | 0.550 | 0.495 | 0.549 | 0.553 | 0.579 | 0.557 | 0.598 | 0.579 | 0.564 | 0.509 | 0.557 | 0.557 |
| | 360 | 0.885 | 0.682 | 0.845 | 0.684 | 1.372 | 0.925 | 0.864 | 0.725 | 0.898 | 0.672 | 1.070 | 0.834 | 1.106 | 0.814 | 0.853 | 0.764 | 0.910 | 0.672 | 0.910 | 0.745 |
| | Avg | 0.711 | 0.595 | 0.701 | 0.609 | 0.984 | 0.728 | 0.713 | 0.633 | 0.724 | 0.583 | 0.809 | 0.694 | 0.843 | 0.685 | 0.725 | 0.672 | 0.737 | 0.590 | 0.733 | 0.651 |
| Sdwpfm2 | 24 | 0.630 | 0.547 | 0.627 | 0.570 | 0.749 | 0.657 | 0.636 | 0.582 | 0.612 | 0.537 | 0.635 | 0.586 | 0.669 | 0.600 | 0.684 | 0.621 | 0.641 | 0.539 | 0.648 | 0.610 |
| | 360 | 0.983 | 0.739 | 0.978 | 0.736 | 1.340 | 0.877 | 0.995 | 0.783 | 1.028 | 0.745 | 0.965 | 0.788 | 1.215 | 0.863 | 0.987 | 0.780 | 1.058 | 0.726 | 1.020 | 0.819 |
| | Avg | 0.807 | 0.643 | 0.803 | 0.653 | 1.044 | 0.767 | 0.816 | 0.682 | 0.820 | 0.641 | 0.800 | 0.687 | 0.942 | 0.732 | 0.836 | 0.701 | 0.850 | 0.632 | 0.834 | 0.714 |
| Sdwpfh1 | 24 | 0.651 | 0.578 | 0.651 | 0.587 | 0.754 | 0.670 | 0.719 | 0.641 | 0.673 | 0.575 | 0.687 | 0.652 | 1.111 | 0.816 | 0.755 | 0.679 | 0.701 | 0.579 | 0.669 | 0.629 |
| | 360 | 0.830 | 0.677 | 0.841 | 0.700 | 0.831 | 0.727 | 0.897 | 0.757 | 0.886 | 0.712 | 0.849 | 0.752 | 1.089 | 0.823 | 0.853 | 0.762 | 0.882 | 0.680 | 0.939 | 0.804 |
| | Avg | 0.741 | 0.628 | 0.746 | 0.643 | 0.793 | 0.698 | 0.808 | 0.699 | 0.779 | 0.644 | 0.768 | 0.702 | 1.100 | 0.820 | 0.804 | 0.720 | 0.791 | 0.630 | 0.804 | 0.717 |
| Sdwpfh2 | 24 | 0.798 | 0.645 | 0.820 | 0.677 | 0.811 | 0.714 | 0.783 | 0.674 | 0.975 | 0.694 | 0.810 | 0.718 | 0.834 | 0.691 | 0.892 | 0.740 | 0.829 | 0.632 | 0.769 | 0.680 |
| | 360 | 0.974 | 0.757 | 0.962 | 0.761 | 1.042 | 0.779 | 1.056 | 0.829 | 1.039 | 0.735 | 1.102 | 0.830 | 1.126 | 0.825 | 0.990 | 0.825 | 1.053 | 0.742 | 1.280 | 0.925 |
| | Avg | 0.886 | 0.701 | 0.891 | 0.719 | 0.926 | 0.746 | 0.919 | 0.751 | 1.007 | 0.715 | 0.956 | 0.774 | 0.980 | 0.768 | 0.941 | 0.782 | 0.941 | 0.687 | 1.025 | 0.802 |
| Colbun | 10 | 0.070 | 0.094 | 0.084 | 0.146 | 0.449 | 0.279 | 0.095 | 0.140 | 0.073 | 0.106 | 0.068 | 0.101 | 0.077 | 0.120 | 0.091 | 0.129 | 0.071 | 0.092 | 0.083 | 0.125 |
| | 30 | 0.168 | 0.236 | 0.181 | 0.293 | 0.663 | 0.494 | 0.282 | 0.341 | 0.221 | 0.290 | 0.190 | 0.289 | 0.227 | 0.301 | 0.311 | 0.377 | 0.405 | 0.379 | 0.217 | 0.316 |
| | Avg | 0.119 | 0.165 | 0.132 | 0.219 | 0.556 | 0.386 | 0.188 | 0.240 | 0.147 | 0.198 | 0.129 | 0.195 | 0.152 | 0.210 | 0.201 | 0.253 | 0.238 | 0.235 | 0.150 | 0.221 |
| Rapel | 10 | 0.217 | 0.221 | 0.301 | 0.308 | 0.238 | 0.274 | 0.238 | 0.270 | 0.221 | 0.225 | 0.201 | 0.224 | 0.240 | 0.259 | 0.226 | 0.238 | 0.421 | 0.421 | 0.215 | 0.232 |
| | 30 | 0.313 | 0.346 | 0.387 | 0.416 | 0.377 | 0.394 | 0.409 | 0.444 | 0.387 | 0.386 | 0.278 | 0.374 | 0.433 | 0.436 | 0.458 | 0.437 | 0.307 | 0.343 | 0.436 | 0.436 |
| | Avg | 0.265 | 0.284 | 0.344 | 0.362 | 0.308 | 0.334 | 0.323 | 0.357 | 0.304 | 0.306 | 0.240 | 0.299 | 0.337 | 0.348 | 0.342 | 0.337 | 0.364 | 0.382 | 0.325 | 0.334 |
| 1st Count | | 18 | 13 | 5 | 5 | 1 | 0 | 0 | 0 | 1 | 6 | 7 | 3 | 0 | 0 | 3 | 2 | 0 | 7 | 1 | 0 |

Table 10: Results under the setting where exogenous variables are partially missing with missing ratios of 10%, 30%, and 50%. The best results are highlighted in **bold**.

| Mask Type | | Zero | | | | | | | | | | Random | | | | | | | | | |
|---|---|---|---|---|---|---|---|---|---|---|---|---|---|---|---|---|---|---|---|---|---|
| Models | | GCGNet | | TimeXer | | CrossLinear | | DUET | | PatchTST | | GCGNet | | TimeXer | | CrossLinear | | DUET | | PatchTST | |
| Metrics | | mse | mae | mse | mae | mse | mae | mse | mae | mse | mae | mse | mae | mse | mae | mse | mae | mse | mae | mse | mae |
| Colbun | 10 | 0.066 | **0.088** | 0.081 | 0.145 | **0.063** | 0.101 | 0.128 | 0.134 | 0.064 | 0.109 | 0.070 | **0.107** | 0.115 | 0.176 | **0.068** | 0.111 | 0.104 | 0.128 | 0.081 | 0.124 |
| | 30 | 0.081 | **0.093** | 0.086 | 0.145 | **0.063** | 0.101 | 0.094 | 0.122 | 0.074 | 0.109 | 0.068 | **0.092** | 0.111 | 0.168 | 0.065 | 0.102 | 0.070 | 0.111 | 0.081 | 0.120 |
| | 50 | 0.081 | **0.096** | 0.084 | 0.147 | 0.070 | 0.105 | 0.096 | 0.121 | 0.071 | 0.107 | 0.073 | **0.090** | 0.110 | 0.164 | 0.065 | 0.111 | 0.078 | 0.115 | 0.074 | 0.114 |
| | avg | 0.076 | **0.093** | 0.084 | 0.145 | **0.065** | 0.102 | 0.106 | 0.126 | 0.070 | 0.108 | 0.070 | **0.096** | 0.112 | 0.169 | **0.066** | 0.108 | 0.084 | 0.118 | 0.078 | 0.120 |
| Energy | 10 | **0.064** | **0.192** | 0.119 | 0.270 | 0.094 | 0.237 | 0.092 | 0.234 | 0.108 | 0.260 | **0.068** | **0.199** | 0.119 | 0.269 | 0.093 | 0.236 | 0.092 | 0.234 | 0.105 | 0.255 |
| | 30 | **0.071** | **0.205** | 0.121 | 0.273 | 0.098 | 0.242 | 0.102 | 0.248 | 0.106 | 0.254 | 0.091 | 0.230 | 0.124 | 0.277 | 0.099 | 0.244 | 0.102 | 0.247 | 0.099 | 0.244 |
| | 50 | **0.077** | **0.212** | 0.125 | 0.277 | 0.100 | 0.244 | 0.106 | 0.252 | 0.108 | 0.256 | 0.085 | 0.223 | 0.128 | 0.280 | 0.101 | 0.246 | 0.107 | 0.253 | 0.103 | 0.248 |
| | avg | **0.070** | **0.203** | 0.122 | 0.273 | 0.097 | 0.241 | 0.100 | 0.244 | 0.108 | 0.257 | **0.081** | **0.217** | 0.123 | 0.275 | 0.098 | 0.242 | 0.100 | 0.245 | 0.102 | 0.249 |
| DE | 10 | **0.276** | **0.325** | 0.391 | 0.387 | 0.410 | 0.406 | 0.398 | 0.391 | 0.452 | 0.433 | **0.286** | **0.338** | 0.385 | 0.383 | 0.417 | 0.408 | 0.413 | 0.395 | 0.461 | 0.435 |
| | 30 | **0.279** | **0.333** | 0.407 | 0.398 | 0.430 | 0.418 | 0.438 | 0.411 | 0.486 | 0.447 | **0.335** | **0.364** | 0.428 | 0.408 | 0.438 | 0.421 | 0.432 | 0.407 | 0.499 | 0.453 |
| | 50 | **0.323** | **0.349** | 0.456 | 0.424 | 0.439 | 0.423 | 0.446 | 0.418 | 0.501 | 0.454 | **0.409** | **0.397** | 0.487 | 0.441 | 0.446 | 0.426 | 0.449 | 0.418 | 0.510 | 0.459 |
| | avg | **0.292** | **0.336** | 0.418 | 0.403 | 0.426 | 0.416 | 0.428 | 0.407 | 0.480 | 0.445 | **0.344** | **0.366** | 0.433 | 0.411 | 0.433 | 0.418 | 0.431 | 0.407 | 0.490 | 0.449 |
| PJM | 10 | **0.058** | **0.147** | 0.090 | 0.186 | 0.096 | 0.200 | 0.076 | 0.172 | 0.111 | 0.233 | **0.061** | **0.151** | 0.090 | 0.187 | 0.095 | 0.197 | 0.075 | 0.171 | 0.107 | 0.221 |
| | 30 | **0.064** | **0.155** | 0.104 | 0.200 | 0.097 | 0.202 | 0.078 | 0.178 | 0.121 | 0.248 | **0.066** | **0.158** | 0.105 | 0.203 | 0.103 | 0.203 | 0.080 | 0.177 | 0.112 | 0.222 |
| | 50 | **0.070** | **0.162** | 0.108 | 0.207 | 0.101 | 0.207 | 0.083 | 0.181 | 0.128 | 0.256 | **0.072** | **0.164** | 0.109 | 0.207 | 0.104 | 0.205 | 0.082 | 0.179 | 0.116 | 0.227 |
| | avg | **0.064** | **0.155** | 0.100 | 0.198 | 0.098 | 0.203 | 0.079 | 0.177 | 0.120 | 0.246 | **0.066** | **0.158** | 0.102 | 0.199 | 0.101 | 0.201 | 0.079 | 0.176 | 0.112 | 0.224 |
| NP | 10 | **0.194** | **0.228** | 0.276 | 0.290 | 0.410 | 0.406 | 0.252 | 0.268 | 0.256 | 0.287 | **0.193** | **0.229** | 0.280 | 0.293 | 0.240 | 0.277 | 0.254 | 0.267 | 0.258 | 0.287 |
| | 30 | **0.200** | **0.233** | 0.285 | 0.296 | 0.430 | 0.418 | 0.259 | 0.270 | 0.270 | 0.295 | **0.206** | **0.234** | 0.296 | 0.296 | 0.249 | 0.287 | 0.261 | 0.271 | 0.271 | 0.292 |
| | 50 | **0.218** | **0.244** | 0.307 | 0.303 | 0.439 | 0.423 | 0.263 | 0.273 | 0.277 | 0.298 | **0.224** | **0.246** | 0.302 | 0.301 | 0.254 | 0.292 | 0.265 | 0.273 | 0.278 | 0.295 |
| | avg | **0.204** | **0.235** | 0.289 | 0.296 | 0.426 | 0.416 | 0.258 | 0.270 | 0.268 | 0.293 | **0.208** | **0.236** | 0.293 | 0.297 | 0.248 | 0.285 | 0.260 | 0.270 | 0.269 | 0.291 |

## THE USE OF LARGE LANGUAGE MODELS

We promise not to use Large Language Models in writing.

