# OpenReview forum: "GCGNet: Graph-Consistent Generative Network for Time Series Forecasting with Exogenous Variables"
_ICLR.cc/2026/Conference — ICLR 2026 Poster_

### Official Review · Reviewer_s4T3 · 2025-10-22

**Soundness:** 4
**Presentation:** 4
**Contribution:** 3
**Rating:** 8
**Confidence:** 5

**Summary:**

This paper introduces GCGNet, a Graph-Consistent Generative Network aimed at time series forecasting with exogenous variables. The framework combines a variational generator for coarse forecasting, a graph-based discriminator enforcing joint temporal and channel consistency, and a graph refiner that leverages learned adjacency structures to refine outputs. Empirical evaluation on 12 real-world datasets, including settings with missing values and ablation studies, demonstrates the proposed method’s improved robustness and competitiveness compared to a suite of baselines.

**Strengths:**

- This paper breaks the existing mindset in forecasting with exogenous variables, which typically follows a two-step modeling strategy inherited from traditional forecasting methods. Instead, it jointly models the time and channel, which represent a effective approach for forecasting with exogenous variables.

- The proposed Graph Discriminator in this paper is interesting, as it provides a novel way to align the predicted values with the underlying temporal and channel joint correlations.

- The experimental results demonstrate that the proposed method achieves strong performance.

- The code and datasets has been open, supporting reproducibility.

**Weaknesses:**

- The description of the Sparsify operation in Section 3.4 only briefly explains its purpose but doesn't  clearly describe how it actually works. More details are needed to help  readers understand what this step does and how it fits into the Graph  Refiner module.

- The Sparsify operation uses a hyperparameter k that controls how much sparsification is applied, and this likely has a big impact on how well the model performs. However, the paper doesn't  explain how the value of k was chosen, or test how different choices of k affect the results.

- Please carefully check the references. References to preprints, such as "Long-term forecasting with tide: Time-series dense encoder. " should be updated to their formally published versions.

**Questions:**

- Please provide a more detailed explanation of the Sparsify operation and conduct a sensitivity analysis on the parameter k  within Sparsify.

---

> ### Author Response · Authors · 2025-11-21
> **Response to Reviewer s4T3**
>
> Thank you very much for your thoughtful and constructive comments. We greatly appreciate that you recognized the main innovation of our work, which breaks the traditional two-step approach commonly used in forecasting with exogenous variables and instead jointly models temporal and channel correlations.
>
> **W1**: Sparsify step is insufficiently explained.
>
> Thank you for your valuable feedback! In the original paper, we only provided a brief introduction to the Sparsify step. To help readers better understand how this operation works, we have added a formal definition of the Sparsify operation $\underline{\text{in lines 278–284 of the revised paper}}$. The formal definition is as follows:
>
> "Formally, given the adjacency matrix $\hat{A} \in \mathbb{R}^{N \times N}$, where $N$ is the number of nodes, we define a sparsity ratio $\alpha \in (0,1)$. For each row, we keep the top $k = N - \lceil \alpha N \rceil$ largest weights and set the remaining entries to zero. The resulting sparse adjacency matrix is denoted by $A_s \in \mathbb{R}^{N \times N}$ and can be expressed as:\begin{align}
> A_s[i,j] =
> \begin{cases}
> \hat{A}[i,j], & \text{if } \hat{A}[i,j] \text{ is among the top-} k \text{ largest values in row } i,\cr
> 0, & \text{otherwise}.
> \end{cases}
> \end{align}"
> **W2**: No analysis of the impact of the Sparsify step hyperparameter k.
>
> Thank you for your suggestion! We apologize for not explaining the setting of the hyperparameter k in the Sparsify step. In our implementation, we define a sparsity ratio $\alpha \in (0,1)$. For each row, we keep the top $k = N - \lceil \alpha N \rceil$ largest weights and set the remaining entries to zero. We conducted a sensitivity analysis on this sparsify ratio, testing 10%, 30%, 50%, 70%, and 90%. The results show that 50% generally yields the best performance across most datasets, as it removes most of the uninformative weights without losing critical information. Therefore, we kept the setting of 50% in our experiments. We have updated the parameter sensitivity experiment $\underline{\text{in lines 855–861 of the revised paper (Appendix B.3)}}$.
> | **Sparsity Ratio** | **0.1** | **0.3** | **0.5** | **0.7** | **0.9** |
> | --- | --- | --- | --- | --- | --- |
> | DE | 0.488 | 0.476 | 0.445 | 0.478 | 0.619 |
> | Energy | 0.333 | 0.370 | 0.330 | 0.381 | 0.409 |
> | NP | 0.497 | 0.523 | 0.442 | 0.496 | 0.592 |
> | PJM | 0.228 | 0.228 | 0.223 | 0.261 | 0.235 |
>
> **W3**: References need updating from preprints to published versions.
>
> Thank you for your helpful feedback. We have carefully reviewed all the references in the paper and updated those that were previously cited as preprints. This ensures that all references now point to the final, peer-reviewed sources, providing readers with accurate and reliable information.
>
> **Q1**: Request detailed explanation of Sparsify and sensitivity analysis on k.
>
> Thank you for your suggestion! As noted above, we have provided a more detailed explanation of the sparsify step (see **W1**) and conducted a sensitivity analysis on sparsify step hyperparameter $k$ (see **W2**).

---

> > ### Comment · Reviewer_s4T3 · 2025-11-26
> >
> > Thanks for your response. Since my score is already positive, I will keep my rating.

---

> > > ### Author Response · Authors · 2025-11-26
> > > **Official Comment by Authors**
> > >
> > > Thank you for your suggestions. We deeply appreciate the opportunity to enhance the quality of our paper. Best wishes!

---

### Official Review · Reviewer_soMm · 2025-10-25

**Soundness:** 3
**Presentation:** 4
**Contribution:** 3
**Rating:** 6
**Confidence:** 4

**Summary:**

The paper proposes GCGNet, a general and robust framework for time series forecasting that incorporates both historical and future exogenous variables. A key component is the Graph Discriminator, which utilizes graph structure consistency to guide the generator, ensuring that the generated forecasts align with temporal and inter-channel dependencies. Extensive experiments are conducted on 12 real-world datasets.

**Strengths:**

1. The paper is well structured, and it focuses on an important problem that could have real-world applications.

2. Nice figures help to understand the proposed method, and both the figures and tables are clear and well-presented.

3. The proposed method may address some limitations of existing approaches, such as separate modeling and limited robustness.

**Weaknesses:**

1. The current set of baselines includes many models that do not inherently accommodate exogenous variables; however, this paper focuses on forecasting with exogenous variables.

2. The adaptation of models that inherently lack support for exogenous variables, such as PatchTST, using a simple MLP fusion approach to incorporate future covariates raises concerns about its effectiveness. It remains questionable whether this method preserves the original modeling principles or potentially introduces degradation in performance.

3. The experimental results for the scenario where future exogenous variables are unavailable are only reported on a subset of datasets.  This is questionable, particularly since this scenario represents a more common and practical use case in real-world applications.

4. The naming of the key module as "Graph Discriminator" strongly suggests an adversarial training paradigm. However, its loss function L_disc functions more as an L1-based graph structure  reconstruction loss rather than a true discriminator performing binary  classification to distinguish "real" from "generated" graphs. This nomenclature is misleading.

**Questions:**

1. Since this paper focuses on forecasting with exogenous variables, why weren’t the baselines restricted to algorithms that natively support exogenous variables?

2. Why not use the original model directly, and instead add an MLP fusion approach, since this module might actually degrade the model’s performance?

3. Why are the experimental results for the scenario without future exogenous variables reported only on a subset of datasets?

4. Why is the key module named "Graph Discriminator" when it is actually unrelated to an adversarial training paradigm?

**Details Of Ethics Concerns:**

None.

---

> ### Author Response · Authors · 2025-11-21
> **Response to Reviewer soMm (Part I)**
>
> Thank you for your thoughtful and constructive comments. We appreciate your careful review and the detailed feedback you provided. Your observations highlight important points regarding the baselines, experimental design, and naming of key modules, which help us improve the clarity and presentation of our paper.
>
> **W1**: Baselines do not all support exogenous variables.
>
> Thank you for your valuable feedback! Since there are relatively few algorithms that natively support exogenous variables, we have evaluated a range of models designed for this setting, including TimeXer, TFT, TiDE, and CrossLinear. To provide a broader comparison, we also tested models that do not natively support exogenous variables. For these models, instead of applying them directly, we employed an MLP fusion module to enable them to effectively utilize exogenous information, which is described in detail $\underline{\text{in lines 745–755 of the revised paper (Appendix B.3)}}$.
>
> **W2**: Effectiveness of MLP fusion module is unclear.
>
> Thank you for your valuable feedback! The original paper did not include a comparison between the original models and their versions augmented with the MLP fusion module in the forecasting scenario with exogenous variables. To demonstrate that our MLP fusion module effectively leverages exogenous variables and improves forecasting performance, we conducted additional experiments. We compared the original models, including DUET, Amplifier, TimeKAN, xPatch, and PatchTST, with their MLP-augmented versions. In addition, we evaluated two other fusion modules designed to enable traditional forecasting models to incorporate future exogenous variables: a convolution-based module and a cross-attention-based module. Specifically, given historical endogenous variables $X^{\text{endo}} \in \mathbb{R}^{N \times T}$ and historical exogenous variables $X^{\text{exo}} \in \mathbb{R}^{D \times T}$, the backbone model first generates a latent representation:
>
> \begin{align}
> z = \theta_{\text{Model}}(X^{\text{endo}}, X^{\text{exo}}),
> \end{align}
>
> where $z \in \mathbb{R}^{N \times F}$ and $\theta_{\text{Model}}$ denotes the parameters of the forecasting model.
>
> The latent representation $z$ is then combined with the future exogenous variables $Y^{\text{exo}} \in \mathbb{R}^{N \times F}$ and processed by either:
>
> 1. **Conv Fusion Module**: a 1D convolution layer captures local correlations across channels, and the resulting feature is fused with the future exogenous variables to produce the final prediction.
> \begin{align}
> \hat{Y}^{\text{endo}} = \theta_{\text{Conv}}\big(\text{Concat}(z, Y^{\text{exo}})\big),
> \end{align}
> where $\theta_{\text{Conv}}$ denotes the parameters of the convolution-based fusion module.
> 2. **Cross-Attention Fusion Module**: a cross-attention layer captures global dependencies between the future exogenous variables and the latent representation to produce the final prediction
> \begin{align}
> \hat{Y}^{\text{endo}} = \theta_{\text{CA}}\big(z, Y^{\text{exo}}\big),
> \end{align}
> where $\theta_{\text{CA}}$ denotes the parameters of the cross-attention fusion module.
>
> The results show that our MLP fusion module consistently improves performance and is an effective method for using exogenous variables. The other two fusion modules were also tested in detail, and the MLP module achieved the best results, so we selected it as the fusion method.
>
> | **DUET** | **origin** |  | **mlp** |  | **conv** |  | **cross** |  |
> | --- | --- | --- | --- | --- | --- | --- | --- | --- |
> | Metrics | mse | mae | mse | mae | mse | mae | mse | mae |
> | NP | 0.444 | 0.382 | **0.411** | 0.408 | 0.430 | **0.381** | 1.633 | 0.996 |
> | PJM | 0.140 | 0.225 | **0.102** | **0.197** | 0.123 | 0.210 | 0.618 | 0.332 |
> | BE | **0.473** | **0.305** | 0.515 | 0.354 | 0.475 | 0.331 | 0.474 | 0.329 |
> | FR | 0.468 | **0.262** | 0.496 | 0.327 | **0.449** | 0.266 | 0.672 | 0.445 |
> | DE | 0.659 | 0.510 | **0.482** | **0.430** | 0.569 | 0.490 | 0.593 | 0.502 |
> | Energy | 0.162 | 0.310 | 0.203 | 0.367 | **0.156** | **0.308** | 0.329 | 0.440 |
> | Sdwpfm1 | 0.724 | 0.583 | **0.599** | **0.570** | 0.703 | 0.572 | 1.603 | 0.832 |
> | Sdwpfm2 | 0.820 | 0.641 | **0.514** | **0.490** | 0.771 | 0.623 | 1.267 | 0.764 |
> | Sdwpfh1 | 0.780 | 0.637 | **0.539** | **0.516** | 0.750 | 0.624 | 1.562 | 0.840 |
> | Sdwpfh2 | 0.920 | 0.691 | **0.647** | **0.566** | 0.888 | 0.680 | 1.685 | 0.881 |
> | Colbun | **0.148** | **0.198** | 0.198 | 0.266 | 0.181 | 0.238 | 1.761 | 1.019 |
> | Rapel | 0.304 | **0.306** | **0.269** | 0.326 | 0.303 | 0.308 | 1.355 | 0.858 |
> | 1st Count | 2 | 4 | **8** | **6** | 2 | 2 | 0 | 0 |

---

> ### Author Response · Authors · 2025-11-21
> **Response to Reviewer soMm (Part II)**
>
> | **CrossLinear** | **origin** |  | **mlp** |  | **conv** |  | **cross** |  |
> | --- | --- | --- | --- | --- | --- | --- | --- | --- |
> | Metrics | mse | mae | mse | mae | mse | mae | mse | mae |
> | NP | 0.450 | 0.393 | **0.404** | **0.384** | 0.441 | 0.393 | 0.917 | 0.749 |
> | PJM | 0.147 | 0.241 | **0.118** | **0.226** | 0.129 | 0.227 | 0.146 | 0.263 |
> | BE | 0.477 | 0.300 | 0.474 | 0.330 | **0.455** | **0.298** | 0.570 | 0.434 |
> | FR | 0.476 | **0.257** | 0.486 | 0.300 | **0.467** | 0.258 | 0.592 | 0.406 |
> | DE | 0.635 | 0.508 | **0.497** | **0.453** | 0.541 | 0.471 | 0.595 | 0.503 |
> | Energy | 0.200 | 0.336 | 0.183 | 0.324 | **0.180** | **0.321** | 0.345 | 0.470 |
> | Sdwpfm1 | 0.809 | 0.694 | **0.417** | **0.478** | 0.631 | 0.614 | 0.710 | 0.681 |
> | Sdwpfm2 | 0.800 | 0.687 | **0.662** | **0.615** | 0.798 | 0.699 | 1.619 | 1.019 |
> | Sdwpfh1 | 0.768 | 0.702 | **0.671** | **0.625** | 0.767 | 0.701 | 1.643 | 1.036 |
> | Sdwpfh2 | 0.956 | 0.774 | **0.620** | **0.596** | 0.905 | 0.740 | 1.443 | 0.955 |
> | Colbun | 0.129 | 0.195 | 0.181 | 0.186 | **0.120** | **0.186** | 0.878 | 0.722 |
> | Rapel | 0.240 | **0.299** | 0.271 | 0.310 | **0.235** | 0.299 | 0.526 | 0.574 |
> | 1st Count | 0 | 2 | **7** | **7** | 5 | 3 | 0 | 0 |
>
> | **Amplifier** | **origin** |  | **mlp** |  | **conv** |  | **cross** |  |
> | --- | --- | --- | --- | --- | --- | --- | --- | --- |
> | Metrics | mse | mae | mse | mae | mse | mae | mse | mae |
> | NP | 0.520 | 0.448 | **0.420** | **0.418** | 0.502 | 0.449 | 1.545 | 1.009 |
> | PJM | 0.152 | 0.245 | 0.137 | 0.246 | **0.124** | **0.226** | 0.137 | 0.251 |
> | BE | **0.502** | **0.325** | 0.559 | 0.413 | 0.548 | 0.412 | 0.729 | 0.551 |
> | FR | 0.494 | **0.286** | 0.554 | 0.408 | **0.481** | 0.292 | 0.505 | 0.340 |
> | DE | 0.712 | 0.548 | **0.473** | **0.441** | 0.659 | 0.526 | 0.852 | 0.635 |
> | Energy | 0.180 | 0.327 | 0.233 | 0.389 | **0.142** | **0.292** | 0.473 | 0.565 |
> | Sdwpfm1 | 0.843 | 0.685 | **0.437** | **0.490** | 0.737 | 0.650 | 0.891 | 0.742 |
> | Sdwpfm2 | 0.942 | 0.732 | **0.491** | **0.512** | 0.749 | 0.652 | 0.930 | 0.761 |
> | Sdwpfh1 | 1.100 | 0.820 | **0.537** | **0.598** | 1.098 | 0.819 | 1.738 | 1.069 |
> | Sdwpfh2 | 0.980 | 0.768 | **0.521** | **0.581** | 0.944 | 0.755 | 1.062 | 0.847 |
> | Colbun | **0.152** | **0.210** | 0.173 | 0.246 | 0.152 | 0.211 | 1.208 | 0.882 |
> | Rapel | 0.337 | 0.348 | **0.257** | **0.321** | 0.331 | 0.342 | 0.846 | 0.729 |
> | 1st Count | 2 | 3 | **7** | **7** | 3 | 2 | 0 | 0 |
>
> | **TimeKAN** | **origin** |  | **mlp** |  | **conv** |  | **cross** |  |
> | --- | --- | --- | --- | --- | --- | --- | --- | --- |
> | Metrics | mse | mae | mse | mae | mse | mae | mse | mae |
> | NP | 0.484 | 0.435 | **0.405** | 0.419 | 0.451 | **0.417** | 1.635 | 1.049 |
> | PJM | 0.175 | 0.270 | **0.139** | 0.262 | 0.153 | **0.254** | 0.388 | 0.490 |
> | BE | 0.488 | **0.315** | 0.548 | 0.407 | **0.468** | 0.324 | 0.511 | 0.369 |
> | FR | 0.491 | 0.276 | 0.547 | 0.374 | **0.476** | **0.272** | 0.720 | 0.513 |
> | DE | 0.669 | 0.526 | **0.473** | **0.445** | 0.534 | 0.469 | 0.514 | 0.450 |
> | Energy | 0.147 | 0.296 | 0.218 | 0.381 | **0.120** | **0.269** | 0.303 | 0.425 |
> | Sdwpfm1 | 0.725 | 0.672 | **0.447** | **0.534** | 0.671 | 0.646 | 1.314 | 0.907 |
> | Sdwpfm2 | 0.836 | 0.701 | **0.497** | **0.564** | 0.773 | 0.673 | 1.376 | 0.921 |
> | Sdwpfh1 | 0.804 | 0.720 | **0.577** | **0.638** | 0.775 | 0.707 | 1.546 | 0.996 |
> | Sdwpfh2 | 0.941 | 0.782 | **0.647** | **0.672** | 0.911 | 0.769 | 1.672 | 1.032 |
> | Colbun | 0.201 | 0.253 | **0.128** | **0.175** | 0.172 | 0.224 | 0.433 | 0.490 |
> | Rapel | 0.342 | 0.337 | **0.249** | **0.311** | 0.342 | 0.347 | 0.427 | 0.460 |
> | 1st Count | 0 | 1 | **9** | **7** | 3 | 4 | 0 | 0 |
>
> | **xPatch** | **origin** |  | **mlp** |  | **conv** |  | **cross** |  |
> | --- | --- | --- | --- | --- | --- | --- | --- | --- |
> | Metrics | mse | mae | mse | mae | mse | mae | mse | mae |
> | NP | 0.487 | 0.399 | **0.378** | **0.370** | 0.446 | 0.388 | 0.460 | 0.405 |
> | PJM | 0.139 | 0.223 | **0.104** | **0.194** | 0.114 | 0.202 | 0.113 | 0.206 |
> | BE | 0.484 | 0.303 | 0.495 | 0.333 | **0.468** | **0.296** | 0.513 | 0.342 |
> | FR | 0.478 | 0.257 | 0.490 | 0.293 | **0.460** | **0.252** | 0.558 | 0.337 |
> | DE | 0.633 | 0.500 | **0.474** | **0.433** | 0.536 | 0.460 | 0.613 | 0.493 |
> | Energy | 0.170 | 0.314 | **0.148** | **0.303** | 0.161 | 0.304 | 0.242 | 0.375 |
> | Sdwpfm1 | 0.737 | 0.590 | **0.630** | **0.568** | 0.723 | 0.586 | 1.605 | 0.845 |
> | Sdwpfm2 | 0.849 | 0.632 | **0.516** | **0.495** | 0.811 | 0.635 | 1.476 | 0.799 |
> | Sdwpfh1 | 0.791 | 0.630 | **0.498** | **0.505** | 0.775 | 0.624 | 1.569 | 0.840 |
> | Sdwpfh2 | 0.941 | 0.687 | **0.610** | **0.551** | 0.926 | 0.680 | 1.676 | 0.872 |
> | Colbun | **0.144** | 0.196 | 0.153 | 0.210 | 0.150 | **0.192** | 0.300 | 0.322 |
> | Rapel | 0.364 | 0.382 | 0.333 | 0.381 | **0.259** | **0.279** | 1.869 | 1.030 |
> | 1st Count | 1 | 0 | **8** | **8** | 3 | 4 | 0 | 0 |

---

> ### Author Response · Authors · 2025-11-21
> **Response to Reviewer soMm (Part III)**
>
> | **PatchTST** | **origin** |  | **mlp** |  | **conv** |  | **cross** |  |
> | --- | --- | --- | --- | --- | --- | --- | --- | --- |
> | Metrics | mse | mae | mse | mae | mse | mae | mse | mae |
> | NP | 0.456 | 0.401 | **0.390** | **0.396** | 0.463 | 0.414 | 1.445 | 0.973 |
> | PJM | 0.147 | 0.242 | 0.133 | 0.259 | **0.132** | **0.227** | 0.240 | 0.360 |
> | BE | 0.485 | **0.309** | 0.577 | 0.432 | **0.475** | 0.320 | 0.908 | 0.635 |
> | FR | 0.470 | 0.264 | 0.588 | 0.410 | **0.461** | **0.254** | 0.812 | 0.563 |
> | DE | 0.696 | 0.527 | **0.501** | **0.455** | 0.600 | 0.498 | 0.919 | 0.644 |
> | Energy | **0.203** | 0.341 | 0.226 | 0.377 | 0.205 | **0.339** | 0.564 | 0.627 |
> | Sdwpfm1 | 0.733 | 0.651 | **0.435** | **0.502** | 0.712 | 0.631 | 0.775 | 0.724 |
> | Sdwpfm2 | 0.834 | 0.714 | **0.510** | **0.547** | 0.853 | 0.686 | 0.853 | 0.750 |
> | Sdwpfh1 | 0.804 | 0.717 | **0.468** | **0.527** | 0.792 | 0.697 | 0.845 | 0.775 |
> | Sdwpfh2 | 1.025 | 0.802 | **0.549** | **0.580** | 0.977 | 0.794 | 0.899 | 0.783 |
> | Colbun | 0.150 | **0.220** | 0.239 | 0.309 | **0.150** | 0.236 | 1.252 | 0.885 |
> | Rapel | 0.325 | 0.334 | **0.269** | **0.332** | 0.348 | 0.341 | 0.791 | 0.688 |
> | 1st Count | 1 | 2 | **7** | **7** | 4 | 3 | 0 | 0 |
>
> **W3**: Results for scenarios without future exogenous variables are limited.
>
> Thank you for your valuable feedback! We have supplemented the experiments to include all datasets and models in the scenario without future exogenous variables. These results are now included in the $\underline{\text{in lines 432–447 of the revised paper}}$. They demonstrate the effectiveness of our model under this more practical and common setting.
>
> | **Models** | **GCGNet** |  | **TimeXer** |  | **TFT** |  | **TiDE** |  | **DUET** |  | **CrossLinear** |  | **Amplifier** |  | **TimeKAN** |  | **xPatch** |  | **PatchTST** |  |
> | --- | --- | --- | --- | --- | --- | --- | --- | --- | --- | --- | --- | --- | --- | --- | --- | --- | --- | --- | --- | --- |
> | Metrics | mse | mae | mse | mae | mse | mae | mse | mae | mse | mae | mse | mae | mse | mae | mse | mae | mse | mae | mse | mae |
> | NP | **0.425** | **0.377** | $\underline{\text{0.440}}$ | 0.383 | 0.647 | 0.488 | 0.467 | 0.416 | 0.444 | $\underline{\text{0.383}}$ | 0.451 | 0.394 | 0.520 | 0.448 | 0.484 | 0.435 | 0.488 | 0.402 | 0.457 | 0.401 |
> | PJM | **0.133** | **0.217** | 0.141 | 0.229 | 0.200 | 0.270 | 0.158 | 0.253 | 0.140 | 0.226 | 0.147 | 0.241 | 0.152 | 0.245 | 0.175 | 0.270 | $\underline{\text{0.139}}$ | $\underline{\text{0.223}}$ | 0.148 | 0.243 |
> | BE | **0.458** | $\underline{\text{0.301}}$ | 0.477 | 0.301 | 0.563 | 0.354 | 0.547 | 0.348 | $\underline{\text{0.473}}$ | 0.305 | 0.477 | **0.300** | 0.502 | 0.325 | 0.488 | 0.315 | 0.483 | 0.302 | 0.485 | 0.309 |
> | FR | **0.428** | **0.245** | $\underline{\text{0.454}}$ | $\underline{\text{0.247}}$ | 0.535 | 0.288 | 0.494 | 0.290 | 0.468 | 0.262 | 0.476 | 0.257 | 0.494 | 0.286 | 0.491 | 0.276 | 0.477 | 0.257 | 0.470 | 0.264 |
> | DE | **0.611** | **0.497** | 0.659 | 0.507 | 0.684 | 0.515 | 0.644 | 0.519 | 0.660 | 0.513 | 0.635 | 0.508 | 0.712 | 0.548 | 0.669 | 0.526 | $\underline{\text{0.633}}$ | $\underline{\text{0.500}}$ | 0.696 | 0.527 |
> | Energy | $\underline{\text{0.150}}$ | $\underline{\text{0.299}}$ | 0.172 | 0.326 | 0.376 | 0.480 | 0.162 | 0.311 | 0.160 | 0.308 | 0.201 | 0.336 | 0.180 | 0.327 | **0.147** | **0.296** | 0.170 | 0.314 | 0.203 | 0.341 |
> | Sdwpfm1 | **0.700** | 0.602 | 0.701 | 0.609 | 0.984 | 0.728 | 0.713 | 0.633 | 0.724 | **0.583** | 0.809 | 0.694 | 0.843 | 0.685 | 0.725 | 0.672 | 0.737 | $\underline{\text{0.590}}$ | 0.733 | 0.651 |
> | Sdwpfm2 | 0.825 | $\underline{\text{0.633}}$ | 0.803 | 0.653 | 1.044 | 0.767 | 0.816 | 0.682 | 0.820 | 0.641 | **0.800** | 0.687 | 0.942 | 0.732 | 0.836 | 0.701 | 0.850 | **0.632** | 0.834 | 0.714 |
> | Sdwpfh1 | $\underline{\text{0.757}}$ | **0.628** | **0.746** | 0.643 | 0.793 | 0.698 | 0.808 | 0.699 | 0.780 | 0.637 | 0.768 | 0.702 | 1.100 | 0.820 | 0.804 | 0.720 | 0.791 | $\underline{\text{0.630}}$ | 0.804 | 0.717 |
> | Sdwpfh2 | **0.882** | $\underline{\text{0.691}}$ | $\underline{\text{0.891}}$ | 0.719 | 0.926 | 0.746 | 0.919 | 0.751 | 0.920 | 0.691 | 0.956 | 0.774 | 0.980 | 0.768 | 0.941 | 0.782 | 0.941 | **0.687** | 1.025 | 0.802 |
> | Colbun | **0.116** | **0.164** | 0.132 | 0.219 | 0.556 | 0.386 | 0.188 | 0.240 | 0.147 | 0.198 | 0.129 | 0.195 | 0.152 | 0.210 | 0.201 | 0.253 | 0.238 | 0.235 | 0.150 | 0.221 |
> | Rapel | $\underline{\text{0.258}}$ | **0.273** | 0.344 | 0.362 | 0.308 | 0.334 | 0.323 | 0.357 | 0.304 | 0.306 | **0.240** | 0.299 | 0.337 | 0.348 | 0.342 | 0.337 | 0.364 | 0.382 | 0.325 | 0.334 |
> | 1st Count | **8** | **7** | 1 | 0 | 0 | 0 | 0 | 0 | 0 | 1 | 2 | 1 | 0 | 0 | 1 | 1 | 0 | $\underline{\text{2}}$ | 0 | 0 |

---

> ### Author Response · Authors · 2025-11-21
> **Response to Reviewer soMm (Part IV)**
>
> **W4**: **Graph Discriminator** naming is misleading.
>
> Thank you for your valuable feedback! We apologize for giving this module the somewhat misleading name **Graph Discriminator**. In the original paper, the module was named as such because it was inspired by ideas from adversarial learning. To make its function clearer to readers, we have renamed the module **Graph Structure Aligner** in the revised paper. This new name better reflects the module’s actual behavior: aligning the model’s predictions with the underlying graph-structured relationships.
>
> **Q1**: Why restrict baselines to models supporting exogenous variables?
>
> Thank you for your valuable feedback! To enable a broader comparison, we test a range of models supporting exogenous variables as possible, including TImexer, TFT, TiDE, and CrossLinear. In addition, to broaden the comparison, we also incorporated models that do not inherently support exogenous variables. Rather than applying them directly, we equipped these models with an MLP fusion module that allows them to effectively integrate exogenous information, following the implementation described $\underline{\text{in lines 745–755 of the revised paper (Appendix B.3)}}$. As shown in **W2**, incorporating exogenous variables through MLP fusion consistently improves their performance.
>
> **Q2**: Why use MLP fusion instead of original models?
>
> Thank you for your valuable feedback! As mentioned in **W2**, the MLP fusion module consistently improves performance compared to the original models. Moreover, among the alternative strategies, the MLP fusion module is also shown to be a comparatively effective approach.
>
> **Q3**: Why report limited results for scenarios without future exogenous variables?
>
> Thank you for your valuable feedback! As mentioned in **W3**, we have supplemented the results to include all models across all datasets in the scenario without future exogenous variables. These results demonstrate that our model maintains strong performance in this more common scenario.
>
> **Q4**: Why is the key module named "Graph Discriminator"?
>
> Thank you for your valuable feedback! In the revised paper, we have renamed the **Graph Discriminator** to **Graph Structure Aligner** to more accurately reflect its actual role. The original name was inspired by adversarial learning, but it differs from a typical discriminator and does not operate as a binary classifier. Therefore, the original term could indeed mislead readers about its actual function. The new name clarifies its behavior: aligning the model’s predictions with the underlying graph-structured relationships.

---

> > ### Comment · Reviewer_soMm · 2025-11-26
> >
> > Thanks to the author’s detailed response, which has addressed most of my concerns. In particular, the experimental supplements and analyses for W2 and W3 are essential for enhancing the quality of this paper and further strengthening its contribution. Additionally, I note that other reviewers have also expressed positive evaluations of this work. Therefore, I still support the acceptance of this paper and have decided to raise my score from 6 to 8.

---

> > > ### Author Response · Authors · 2025-11-26
> > > **Official Comment by Authors**
> > >
> > > Thank you very much for appreciating our work. Best wishes!

---

### Official Review · Reviewer_wNXH · 2025-10-27

**Soundness:** 3
**Presentation:** 3
**Contribution:** 4
**Rating:** 6
**Confidence:** 5

**Summary:**

In summary, this paper introduces GCGNet, a graph-consistent generative network for time series forecasting with exogenous variables. GCGNet first generates coarse predictions, then enforces structural alignment to capture joint temporal and channel correlations, and finally refines the predictions to prevent degeneration and improve accuracy. With these designs, GCGNet effectively models complex temporal–channel dependencies while maintaining robustness in real-world scenarios.

**Strengths:**

S1. Time series forecasting with exogenous variables is important to various domains.

S2: This paper proposes GCGNet to address the problem of forecasting with exogenous variables.

S3: This paper proposes Graph Discriminator module to ensuring the outputs align with the underlying temporal and channel joint correlation.

S4: The authors released an anonymous GitHub repository, which improves reproducibility.

**Weaknesses:**

W1: In Figure (c), does it correspond to the framework proposed in this paper? \(Y^{endo}\) represents future endogenous variables, so why is \(Y^{endo}\) assumed to be accessible in the framework?

W2: In Figure (c), I noticed that the figure labels it as \(\tilde{Y}^{endo}\), but the figure does not provide any explanation of what \(\tilde{Y}^{endo}\) represents.

W3: The description in lines 240–244 is unclear. In Equation (7), the process of obtaining \(\hat{A}\) from \(A\) is too vague. Since \(A\) represents a graph, the authors should clarify how a graph is processed using the VAE, rather than simply stating that \(\hat{A}\) is obtained from \(A\).

W4: In lines 77–84, the authors claim that using a generative network helps improve robustness. However, they only provide the results in Table 3 to validate the overall framework, without isolating the contribution of the generative network itself. It would be more convincing if the authors could provide theoretical justification or additional experiments, such as an ablation study, to demonstrate that the generative network specifically contributes to the performance improvement.

W5: In lines 428–429, the authors mention that in Table 4 the "random" setting replaces masked values with random values sampled from the normal distribution N(0,1). This is unreasonable because the range and scale of the original features may differ significantly from N(0,1). Such noise does not reflect realistic scenarios, and therefore the "random" experiment is not meaningful.

**Questions:**

See weakness

---

> ### Author Response · Authors · 2025-11-21
> **Response to Reviewer wNXH**
>
> Thank you for your thoughtful and constructive comments.
>
> **W1**: Figure 2 (c) assumes future endogenous variables are accessible, which is unclear.
>
> Thank you for your valuable feedback! We realize that Figure 2(c) was not clearly explained and could potentially mislead readers into thinking that $\tilde{Y}^{endo}$ represents the true future values. In fact, $\tilde{Y}^{endo}$ are coarse predictions of the future endogenous variables generated by the Variational Generator module of GCGNet. To convey the meaning of the figure more clearly, we have revised its caption $\underline{\text{in lines 87–94 of the revised paper}}$. The updated description is as follows:
>
> "Categories of time series forecasting with exogenous variables algorithms: (a) modeling temporal correlations first, then modeling channel correlations; (b) modeling channel correlations first, then modeling temporal correlations; (c) jointly modeling temporal and channel correlations, where $\tilde{Y}^{endo}$ represents a coarse prediction of $Y^{endo}$ detailed in Section 3.2."
>
> **W2**: Figure 2 (c) label $\tilde{Y}^{endo}$ is unexplained.
>
> Thank you for your valuable feedback! As noted in W1, $\tilde{Y}^{endo}$ represents coarse predictions of future endogenous variables generated by the Variational Generator module of GCGNet, rather than the true future values. $\underline{\text{In lines 87–94 of the revised paper}}$, we have updated the caption of Figure 2(c) to clearly explain the meaning of $\tilde{Y}^{endo}$.
>
> **W3**: Process of obtaining $\hat{A}$ from $\tilde{A}$ via VAE is vague.
>
> Thank you for your valuable feedback. The procedure for obtaining $\hat{A}$ with the VAE was not described in detail in the original paper because the steps are straightforward. Our method follows the approach used by Simonovsky and Komodakis (2018)[1] for handling graph in VAEs.
>
> Because $\tilde{A}$ is a symmetric matrix, we only keep its upper triangular part and flatten it into a one-dimensional vector. This vector is used as the input to the VAE encoder, which maps it into a latent representation. The decoder then outputs a reconstructed vector with the same length. We reshape this vector back into an upper triangular matrix and mirror it across the diagonal to recover a full symmetric matrix. The reconstructed symmetric matrix is taken as $\hat{A}$.
>
> [1] Martin Simonovsky and Nikos Komodakis. Graphvae: Towards generation of small graphs using
> variational autoencoders. In ICANN, volume 11139, pp. 412–422, 2018.
>
> **W4**: Contribution of the generative network is not clear.
>
> Thank you for your valuable feedback! To clarify the contribution of the generative network, we conducted additional experiments. We tested the original GCGNet and a comparison model in which all VAE modules were replaced with MLPs (denoted as w/o VAE in the table), under two different masking settings, where **Zero** replaces masked values with 0 and **Random** replaces masked values with values sampled from $\mathcal{N}(0,1)$, consistent with the **Robustness under Missing Values** experiments in the original paper. The results show that GCGNet achieves better performance on noisy datasets, demonstrating that the generative network meaningfully contributes to the model’s robustness and overall effectiveness. This experiment is added $\underline{\text{in lines 473–487 of the revised paper}}$.
>
> | **Mask Type** | **Zero** |  |  |  | **Random** |  |  |  |
> | --- | --- | --- | --- | --- | --- | --- | --- | --- |
> | Models | GCGNet |  | w/o VAE |  | GCGNet |  | w/o VAE |  |
> | Colbun | **0.076** | **0.093** | 0.094 | 0.140 | **0.070** | **0.096** | 0.081 | 0.123 |
> | Energy | **0.070** | **0.203** | 0.142 | 0.293 | **0.081** | **0.217** | 0.142 | 0.291 |
> | DE | **0.292** | **0.336** | 0.453 | 0.430 | **0.344** | **0.366** | 0.505 | 0.460 |
> | PJM | **0.064** | **0.155** | 0.108 | 0.208 | **0.066** | **0.158** | 0.094 | 0.193 |
> | NP | **0.204** | **0.235** | 0.301 | 0.310 | **0.208** | **0.236** | 0.272 | 0.283 |
>
> **W5**: Robustness under Missing Values experiment using N(0,1) noise is unsuitable.
>
> Thank you for your valuable feedback! We apologize for the unclear explanation. In fact, before filling the masked values with N(0,1) noise, the data were normalized to N(0,1). This filling strategy follows the experimental settings used in CrossLinear[1] and TimeXer[2].
>
> [1] Pengfei Zhou, Yunlong Liu, Junli Liang, Qi Song, and Xiangyang Li. Crosslinear: Plug-and-play cross-correlation embedding for time series forecasting with exogenous variables. In SIGKDD, 2025.
>
> [2] Yuxuan Wang, Haixu Wu, Jiaxiang Dong, Guo Qin, Haoran Zhang, Yong Liu, Yunzhong Qiu, Jianmin Wang, and Mingsheng Long. Timexer: Empowering transformers for time series forecasting with exogenous variables. In NeurIPS, volume 37, pp. 469–498, 2024.

---

### Official Review · Reviewer_vFgB · 2025-10-31

**Soundness:** 2
**Presentation:** 2
**Contribution:** 2
**Rating:** 2
**Confidence:** 4

**Summary:**

The paper **proposes GCGNet**, a *Graph-Consistent Generative Network* that jointly models temporal and channel correlations for robust time series forecasting with exogenous variables. Through a variational generator, graph discriminator, and refiner modules, **GCGNet achieves state-of-the-art performance** across 12 real-world datasets, outperforming prior two-step modeling approaches by effectively capturing joint dependencies and handling noisy data.

**Strengths:**

**Strengths:**

1. The experimental results are significant.
2. The writing is clear and easy to follow.

**Weaknesses:**

**Weaknesses:**

1. **Lack of Novelty:** I fail to see significant advantages of this paper over related works. On one hand, graph-based methods for modeling the relationships between variables and the joint modeling of time and variables have been discussed in early literature, such as GWNe[1], DCRNN[2], and STEP[2]. On the other hand, the lack of joint modeling of temporal and channel correlations is not supported by quantitative case analyses. Furthermore, the authors argue that noise affects the model, which motivates the use of generative models, but this lacks persuasiveness. Even generative models are data-driven and cannot fully address the impact of data noise. Overall, the authors should provide more evidence to support their motivation, which is the foundation for the proposed methods and contributions.

2. **Lack of Technical Innovation:** The authors lack a thorough analysis of the challenges encountered while addressing issues in related works. Instead, they directly propose the graph-consistent generative module, including the Variational Generator, Graph Discriminator, Graph Variational Autoencoder, and Graph Refiner modules, without sufficiently justifying each design choice. The overall model design, as well as the design of each part, lacks enough motivation and insights, making the approach feel overly engineered.

3. **Writing Needs Improvement:** As mentioned above, the writing requires significant improvement. The authors should provide more evidence to validate their motivations, the significance of their contributions, and the insights behind the method design.

[1]. Graph WaveNet for Deep Spatial-Temporal Graph Modeling
[2]. Diffusion Convolutional Recurrent Neural Network: Data-Driven Traffic Forecasting
[3]. Pre-training Enhanced Spatial-temporal Graph Neural Network for Multivariate Time Series Forecasting

**Questions:**

See weaknesses.

---

> ### Author Response · Authors · 2025-11-21
> **Response to Reviewer vFgB (Part I)**
>
> Thank you for your thoughtful and constructive comments.
>
> **W1**: Lack of Novelty.
> Thank you for your valuable feedback. We will now address your concerns by providing the following clarifications.
>
> **Joint modeling approach lacks novelity**
>
> Our GCGNet is fundamentally different from these methods. These differences become clear when considering the operational context of the cited models: **DCRNN**, **Graph WaveNet**, and **STEP**. **DCRNN** models the road network as a directed graph and captures spatial correlations through a bidirectional diffusion convolution—essentially a random-walk-based filter that learns separate upstream/downstream influence weights. **Graph WaveNet** learns hidden spatial correlations in traffic networks via an adaptive adjacency matrix discovered from node embeddings, freeing the model from fixed, possibly incomplete road graphs. Stacked dilated causal convolutions then capture long-range temporal patterns efficiently. **STEP** pre-trains a Transformer on long time-series, then uses the learned segment embeddings to build a sparse kNN-regularized graph, giving the downstream STGNN a reliable, data-driven correlation structure.
>
> While our GCGNet also leverages graph learning, there are several fundamental differences:
>
> - **Multivariate Time Series forecasting vs. Covaraite Time Series Forecasting:** A fundamental difference is that our model is designed for forecasting with exogenous variables, whereas the mentioned models focus on standard multivariate forecasting. These models primarily predict the future based on historical data, following a “history → future” information flow. In contrast, our setting assumes that future exogenous variables are known, introducing a new and critical information flow: “future exogenous → future endogenous.” Existing architectures are not designed to capture or leverage such correlations at future time steps.
> - **Channel Graph vs. Temporal-Channel** **Graph:** Due to this different scenario, our implementation differs significantly from prior works. In DCRNN, Graph WaveNet, and STEP, the constructed graphs—whether based on spatial distance or data-driven correlations—represent channel correlations . However, in our scenario, the target endogenous variables depend not only on the historial endogenous variables but also on future exogenous variables. Therefore, our graph represents joint temporal-channel correlations , rather than a channel-based graph.
> - **Separated Modeling vs. Joint Modeling:** In DCRNN, Graph WaveNet, and STEP, joint modeling refers to modeling both temporal and channel correlations sequentially or hierarchically. **DCRNN** use graph to model channel correlations and rnn for temporal dependencies, **Graph WaveNet** use graph to model channel correlations and tcn based wavenet for temporal depednencies. **STEP** generates a graph that helps STGNNs better model the underlying dependencies. In contrast, GCGNet models temporal and channel correlations simultaneously within the graph, enabling a more effective representation of the correlations.
>
> **Effectiveness of joint modeling is unclear**
>
> We apologize for the insufficient demonstration of the effectiveness of joint modeling in the original paper. In fact, the result visualization experiment presented in the original work was specifically designed to validate this effectiveness. Therefore, to more clearly illustrate the advantages of the joint modeling strategy over the two-step strategy, we supplemented **The Advantages of Joint Modeling** $\underline{\text{in lines 489–529 of the revised paper}}$. We supplemented an additional two-step modeling baseline, namely CrossLinear, which first models inter-channel relationships followed by temporal correlations. These results demonstrate the effectiveness of GCGNet’s joint modeling strategy in forecasting with exogenous variables.

---

> ### Author Response · Authors · 2025-11-21
> **Response to Reviewer vFgB (Part II)**
>
> As reflected in the experimental results: when the historical endogenous data exhibit clear periodic patterns, PatchTST, unable to consider exogenous variables, mainly replicates these patterns. Meanwhile, CrossLinear, which models channel correlations first and then temporal correlations, clearly preserves the periodic information. In contrast, GCGNet generates predictions that closely match the ground truth. This demonstrates that GCGNet effectively uses both endogenous and historical exogenous variables by jointly modeling temporal and channel correlations. When future exogenous variables are included, GCGNet’s performance improves further. While an MLP fusion module is applied to PatchTST, enabling it to incorporate future exogenous variables in a two-step modeling strategy. The first step models temporal correlations, and the second step models channel correlations. We observe that this approach provides only limited improvement, as its predictions closely follow the shape of the future grid load, showing that correlations learned in the second step can override those from the first step. A similar behavior can also be observed for CrossLinear. These results demonstrate the effectiveness of GCGNet’s joint modeling strategy in forecasting with exogenous variables, which strongly supports the research motivation of this work.
>
> **Generative models' robustness to noise is unproven**
>
> To clarify the generative models’ ability to handle data noise, we conducted additional experiments comparing the original GCGNet with a variant where all VAE modules were replaced by MLPs, under two masking settings, where **Zero** replaces masked values with 0 and **Random** replaces masked values with values sampled from $\mathcal{N}(0,1)$, consistent with the **Robustness under Missing Values** experiments in the original paper.
>
> | **Mask Type** | **Zero** |  |  |  | **Random** |  |  |  |
> | --- | --- | --- | --- | --- | --- | --- | --- | --- |
> | Models | GCGNet |  | w/o VAE  |  | GCGNet |  | w/o VAE |  |
> | Colbun | **0.076** | **0.093** | 0.094 | 0.140 | **0.070** | **0.096** | 0.081 | 0.123 |
> | Energy | **0.070** | **0.203** | 0.142 | 0.293 | **0.081** | **0.217** | 0.142 | 0.291 |
> | DE | **0.292** | **0.336** | 0.453 | 0.430 | **0.344** | **0.366** | 0.505 | 0.460 |
> | PJM | **0.064** | **0.155** | 0.108 | 0.208 | **0.066** | **0.158** | 0.094 | 0.193 |
> | NP | **0.204** | **0.235** | 0.301 | 0.310 | **0.208** | **0.236** | 0.272 | 0.283 |
>
> The results show that GCGNet consistently outperforms the MLP-based variant on noisy datasets, demonstrating that the generative network significantly enhances the model’s robustness and overall effectiveness. These results validate our motivation.
>
> **W2**: Lack of Technical Innovation.
>
> Thank you for your valuable feedback. The design of these modules is well-motivated.
>
> Our work mainly aims to address the following two challenges:
>
> **Limitations of two-step strategy:** Many existing forecasting methods fail to capture these correlations effectively because they adopt a two-step modeling strategy. These approaches either first model temporal correlations and then channel correlations, or vice versa. This sequential process, however, is a key limitation: the two-step strategy can cause interference between the two stages, thereby limiting the model’s capacity to fully capture the true temporal and channel relationships, which ultimately leads to suboptimal performance.
>
> **Noisy observations in real-world data:** Real-world data is often compromised by sensor failures, transmission errors, and manual mistakes, introducing diverse noise into the observations. This noise significantly complicates the forecasting task, as observed data may not accurately reflect the true underlying correlations. Consequently, conventional models tend to overfit these noisy observations, making it difficult for them to learn reliable, stable correlations.
>
> **To address the limitations of two-step strategy,** we choose graph-based approaches. Graph-based approaches are inherently well-suited for modeling complex relationships by capturing the connections between nodes. This capability enables the joint modeling of temporal and channel correlations simultaneously, providing a potential solution to the limitations of the two-step approaches.
>
> **To address the challenge of noisy observations,** we choose generative models. Generative models offer an advantage: they are designed to learn the underlying data distribution and latent structures, rather than inferring correlations directly from potentially misleading noisy observations, which can enhance robustness. This motivates our overall framework, which integrates a graph-based module with a generative network to jointly model temporal and channel dependencies for robust forecasting.

---

> ### Author Response · Authors · 2025-11-21
> **Response to Reviewer vFgB (Part III)**
>
> To help readers more clearly understand our overall objective, we added a sentence $\underline{\text{in lines 160–161 of the revised paper}}$ summarizing the goal of our framework: “In this work, we aim to model the joint temporal and channel correlations in a robust manner by integrating graph-based and generative models.“
>
> Within this overall framework, each module is carefully designed to address a specific challenge. The **Variational Generator** module  produces coarse intermediate predictions, which is a natural approach when aiming to directly model the dependencies with the target future values. The **Graph Structure Aligner** module (formerly referred to as the Graph Discriminator) is crucial for explicitly modeling the joint temporal and channel dependencies. The **Graph VAE** module ensures that these learned joint dependencies are stable and robust, preventing instability due to noise or limited data. Finally, the **Graph Refiner** module prevents the collapse of the learned dependencies, maintaining the integrity of the overall modeling process. Collectively, these modules work in a complementary manner, and their inclusion is fully motivated by the challenges inherent in forecasting with exogenous variables.
>
> We also conducted ablation studies on the corresponding modules to demonstrate the effectiveness and rationality of each module design.
>
> | **Dataset** | **NP** | **PJM** | **DE** | **Energy** | **Average** |
> | --- | --- | --- | --- | --- | --- |
> | Metrics | mse | mae | mse | mae | mse |
> | (a) Replace Variational Generator | 0.537 | 0.464 | 0.140 | 0.237 | 0.548 |
> | (b) Remove $L_{align}$ | 0.659 | 0.526 | 0.137 | 0.230 | 0.791 |
> | (c) Replace Graph VAE | 0.691 | 0.540 | 0.213 | 0.290 | 0.665 |
> | (d) Remove Graph Refiner | 0.853 | 0.599 | 0.320 | 0.387 | 0.970 |
> | GCGNet | **0.496** | **0.223** | **0.442** | **0.128** | **0.491** |
>
> Replacing the VAE with an MLP degrades performance, which demonstrates the necessity of variational modeling in capturing the uncertainty and diversity of data.
>
> When the $L_{align}$ term is removed, the performance decreases because the Graph Structure Aligner module (formerly referred to as the Graph Discriminator) no longer provides structural guidance. This indicates that $L_{align}$ is crucial for encouraging the Variational Generator module to produce coarse forecasts that are consistent with the underlying correlations.
>
> Replacing the Graph VAE with a Graph Learner, which directly adopts the adjacency matrix produced by the Graph Learner, performs worse. Unlike the Graph VAE, the Graph Learner generates deterministic graphs and cannot capture the uncertainty and diversity of possible graph structures. Consequently, the learned graphs are less expressive and less robust, limiting the model’s ability to represent complex temporal and channel correlations.
>
> Removing the Graph Refiner module results in a significant drop in performance, since the Graph Structure Aligner module (formerly referred to as the Graph Discriminator) is applied but its generated adjacency matrix is not used, the Graph VAE tends to degenerate into producing meaningless outputs, which in turn causes model collapse. The Graph Refiner module prevents this by incorporating $\tilde{A}$ into the prediction process, thereby forcing the VAE to produce informative adjacency matrices and enabling the model to capture temporal and channel correlations.
>
> These results of ablation studies support the motivation for GCGNet.

---

> ### Author Response · Authors · 2025-11-21
> **Response to Reviewer vFgB (Part IV)**
>
> **W3**: Writing Needs Improvement.
>
> Thank you for your valuable feedback. We acknowledge that some parts of our original paper did not sufficiently provide evidence for the motivation behind our design choices and that the writing regarding motivation needed improvement. To address this, we have made the following modifications:
>
> 1. To justify the use of the generative network, we added an experiment $\underline{\text{in lines 472–488 of the revised paper}}$ that demonstrates the effectiveness of our design.
> 2. To support the motivation for joint modeling, we modified the original visualization experiment into a dedicated motivation experiment $\underline{\text{in lines 489–530 of the revised paper}}$, showing the advantages of joint modeling over a two-stage approach.
> 3. We revised the methodology section to ensure that the design motivation of each module is clearly explained. Motivational statements have been added before each module description to guide the reader more clearly:
>     - **Variational Generator**: $\underline{\text{In lines 160–161 of the revised paper}}$, we added the following motivation statement:
>
>         "The Variational Generator module aims to produce a coarse generation of the future sequence, facilitating the subsequent modeling of dependencies."
>
>     - **Graph Structure Aligner**: $\underline{\text{In lines 202–204 of the revised paper}}$, we added the following motivation statement:
>
>         "The Graph Structure Aligner module is introduced to constrain the generative process of the Variational Generator module through the alignment of correlations."
>
>     - **Graph Refiner**: The original paper already provides a detailed statement of the motivation. Therefore, we have not made changes; the relevant content can be found $\underline{\text{in lines 266–272 of the revised paper}}$.

---

> > ### Comment · Reviewer_vFgB · 2025-11-23
> >
> > Thank you for your thorough response. I have carefully read it and believe that my concerns have been fully addressed; therefore, I will increase my score for the paper. In addition, I think the work would be further strengthened by including broader experimental comparisons, such as recent graph-based time series forecasting models like TimeFilter [1] and MSGNet [2], which would enhance the persuasiveness and overall quality of the paper.
> >
> >
> > [1] TimeFilter: Patch-specific spatial-temporal graph filtration for time series forecasting
> >
> > [2] MSGNet: Learning Multi-Scale Inter-Series Correlations for Multivariate Time Series Forecasting

---

> > > ### Author Response · Authors · 2025-11-27
> > > **Official Comment by Authors (Part I)**
> > >
> > > Thank you for your recognition of our work and your valuable feedback! The reason we did not include other graph-based methods is that, to the best of our knowledge, no existing graph-based forecasting models are designed for forecasting with exogenous variables.
> > >
> > > Following your suggestion, we added comparisons with TimeFilter and MSGNet. To make the comparison more comprehensive, we considered two settings: one where future exogenous variables are available and another where future exogenous variables are unavailable.
> > >
> > > The following results are obtained under the setting where future exogenous variables are available. For a fair comparison, we equip TimeFilter and MSGNet with the MLP fusion approach (see $\underline{\text{lines 745–755 of the revised paper (Appendix B.3)}}$), enabling them to incorporate future exogenous variables. The results indicate that GCGNet still achieves superior performance.
> > >
> > > |Models||GCGNet||TimeFilter||MSGNet||
> > > |---|---|---|---|---|---|---|---|
> > > |Metrics||mse|mae|mse|mae|mse|mae|
> > > |NP|24|**0.197**|**0.233**|0.243|0.296| $\underline{\text{0.200}}$ |$\underline{\text{0.259}}$|
> > > ||360|$\underline{\text{0.496}}$|**0.442**|**0.478**|$\underline{\text{0.474}}$|0.625|0.539|
> > > ||Avg|**0.346**|**0.337**|$\underline{\text{0.361}}$|$\underline{\text{0.385}}$|0.413|0.399|
> > > |PJM|24|**0.058**|**0.148**|0.154|0.283|$\underline{\text{0.089}}$|$\underline{\text{0.206}}$|
> > > ||360|**0.128**|**0.223**|$\underline{\text{0.142}}$|$\underline{\text{0.279}}$|0.157|0.284|
> > > ||Avg|**0.093**|**0.186**|0.148|0.281|$\underline{\text{0.123}}$|0$\underline{\text{.245}}$|
> > > |BE|24|**0.323**|**0.227**|0.448|0.314|$\underline{\text{0.436}}$|$\underline{\text{0.297}}$|
> > > ||360|**0.524**|**0.347**|0.737|0.565|$\underline{\text{0.651}}$|$\underline{\text{0.498}}$|
> > > ||Avg|**0.423**|**0.287**|0.593|0.439|$\underline{\text{0.543}}$|$\underline{\text{0.397}}$|
> > > |FR|24|**0.332**|**0.179**|$\underline{\text{0.448}}$|$\underline{\text{0.308}}$|0.469|0.338|
> > > ||360|**0.478**|**0.280**|$\underline{\text{0.673}}$|$\underline{\text{0.494}}$|0.747|0.496|
> > > ||Avg|**0.405**|**0.230**|$\underline{\text{0.561}}$|$\underline{\text{0.401}}$|0.608|0.417|
> > > |DE|24|**0.282**|**0.328**|$\underline{\text{0.388}}$|$\underline{\text{0.399}}$|0.418|0.402|
> > > ||360|**0.491**|**0.445**|0.576|$\underline{\text{0.483}}$|$\underline{\text{0.575}}$|0.488|
> > > ||Avg|**0.387**|**0.387**|$\underline{\text{0.482}}$|$\underline{\text{0.441}}$|0.496|0.445|
> > > |Energy|24|**0.065**|**0.194**|$\underline{\text{0.279}}$|$\underline{\text{0.423}}$|0.385|0.545|
> > > ||360|**0.179**|**0.330**|0.455|0.581|$\underline{\text{0.313}}$|$\underline{\text{0.453}}$|
> > > ||Avg|**0.122**|**0.262**|0.367|0.502|$\underline{\text{0.349}}$|$\underline{\text{0.499}}$|
> > > |Sdwpfm1|24|**0.375**|**0.409**|$\underline{\text{0.386}}$|$\underline{\text{0.478}}$|0.417|0.492|
> > > ||360|**0.458**|**0.490**|$\underline{\text{0.494}}$|$\underline{\text{0.531}}$|0.718|0.624|
> > > ||Avg|**0.416**|**0.449**|$\underline{\text{0.440}}$|$\underline{\text{0.504}}$|0.567|0.558|
> > > |Sdwpfm2|24|$\underline{\text{0.410}}$|**0.443**|0.410|0.483|**0.388**|$\underline{\text{0.478}}$|
> > > ||360|**0.505**|**0.514**|0.628|0.580|$\underline{\text{0.595}}$|$\underline{\text{0.570}}$|
> > > ||Avg|**0.458**|**0.479**|0.519|0.531|$\underline{\text{0.492}}$|$\underline{\text{0.524}}$|
> > > |Sdwpfh1|24|$\underline{\text{0.381}}$|**0.441**|0.413|0.490|**0.378**|$\underline{\text{0.462}}$|
> > > ||360|**0.446**|**0.495**|$\underline{\text{0.473}}$|0.542|0.526|$\underline{\text{0.526}}$|
> > > ||Avg|**0.414**|**0.468**|$\underline{\text{0.443}}$|0.516|0.452|$\underline{\text{0.494}}$|
> > > |Sdwpfh2|24|**0.426**|**0.474**|0.473|0.536|$\underline{\text{0.432}}$|$\underline{\text{0.493}}$|
> > > ||360|**0.519**|**0.529**|$\underline{\text{0.599}}$|0.626|0.623|$\underline{\text{0.574}}$|
> > > ||Avg|**0.472**|**0.501**|0.536|0.581|$\underline{\text{0.528}}$|$\underline{\text{0.534}}$|
> > > |Colbun|10|**0.057**|**0.084**|0.313|0.419|$\underline{\text{0.097}}$|$\underline{\text{0.159}}$|
> > > ||30|**0.138**|**0.224**|$\underline{\text{0.499}}$|0.514|0.563|$\underline{\text{0.498}}$|
> > > ||Avg|**0.098**|**0.154**|0.406|0.466|$\underline{\text{0.330}}$|$\underline{\text{0.329}}$|
> > > |Rapel|10|**0.140**|**0.191**|0.200|$\underline{\text{0.235}}$|$\underline{\text{0.194}}$|0.258|
> > > ||30|**0.284**|**0.327**|0.406|0.440|$\underline{\text{0.321}}$|$\underline{\text{0.428}}$|
> > > ||Avg|**0.212**|**0.259**|0.303|$\underline{\text{0.337}}$|$\underline{\text{0.257}}$|0.343|

---

> > > > ### Author Response · Authors · 2025-11-27
> > > > **Official Comment by Authors (Part II)**
> > > >
> > > > The following results are obtained under the setting where future exogenous variables are not available. The results indicate that GCGNet still achieves superior performance.
> > > >
> > > > | Models |  | GCGNet |  | TimeFilter |  | MSGNet |  |
> > > > | --- | --- | --- | --- | --- | --- | --- | --- |
> > > > | Metrics |  | mse | mae | mse | mae | mse | mae |
> > > > | NP | 24 | $\underline{\text{0.234}}$ | **0.259** | 0.278 | 0.302 | **0.231** | $\underline{\text{0.271}}$ |
> > > > |  | 360 | **0.617** | **0.495** | $\underline{\text{0.671}}$ | $\underline{\text{0.532}}$ | 0.751 | 0.581 |
> > > > |  | Avg | **0.425** | **0.377** | $\underline{\text{0.475}}$ | $\underline{\text{0.417}}$ | 0.491 | 0.426 |
> > > > | PJM | 24 | **0.073** | **0.168** | 0.113 | 0.221 | $\underline{\text{0.086}}$ | $\underline{\text{0.188}}$ |
> > > > |  | 360 | **0.192** | **0.266** | $\underline{\text{0.207}}$ | $\underline{\text{0.295}}$ | 0.242 | 0.306 |
> > > > |  | Avg | **0.133** | **0.217** | $\underline{\text{0.160}}$ | 0.258 | 0.164 | $\underline{\text{0.247}}$ |
> > > > | BE | 24 | **0.363** | **0.251** | $\underline{\text{0.393}}$ | $\underline{\text{0.264}}$ | 0.395 | 0.271 |
> > > > |  | 360 | **0.553** | **0.351** | $\underline{\text{0.636}}$ | $\underline{\text{0.390}}$ | 0.683 | 0.419 |
> > > > |  | Avg | **0.458** | **0.301** | $\underline{\text{0.515}}$ | $\underline{\text{0.327}}$ | 0.539 | 0.345 |
> > > > | FR | 24 | **0.338** | **0.194** | $\underline{\text{0.401}}$ | $\underline{\text{0.230}}$ | 0.490 | 0.244 |
> > > > |  | 360 | **0.519** | **0.296** | $\underline{\text{0.573}}$ | $\underline{\text{0.325}}$ | 0.663 | 0.367 |
> > > > |  | Avg | **0.428** | **0.245** | $\underline{\text{0.487}}$ | $\underline{\text{0.278}}$ | 0.576 | 0.305 |
> > > > | DE | 24 | **0.415** | **0.400** | 0.446 | 0.425 | $\underline{\text{0.433}}$ | $\underline{\text{0.411}}$ |
> > > > |  | 360 | **0.808** | **0.594** | $\underline{\text{0.847}}$ | $\underline{\text{0.596}}$ | 0.912 | 0.636 |
> > > > |  | Avg | **0.611** | **0.497** | $\underline{\text{0.647}}$ | $\underline{\text{0.511}}$ | 0.672 | 0.523 |
> > > > | Energy | 24 | **0.102** | **0.246** | $\underline{\text{0.113}}$ | $\underline{\text{0.259}}$ | 0.122 | 0.270 |
> > > > |  | 360 | **0.197** | **0.352** | 0.457 | 0.527 | $\underline{\text{0.266}}$ | $\underline{\text{0.414}}$ |
> > > > |  | Avg | **0.150** | **0.299** | 0.285 | 0.393 | $\underline{\text{0.194}}$ | $\underline{\text{0.342}}$ |
> > > > | Sdwpfm1 | 24 | **0.536** | **0.493** | $\underline{\text{0.586}}$ | $\underline{\text{0.553}}$ | 0.605 | 0.568 |
> > > > |  | 360 | **0.864** | **0.712** | $\underline{\text{1.048}}$ | $\underline{\text{0.816}}$ | 1.081 | 0.822 |
> > > > |  | Avg | **0.700** | **0.602** | $\underline{\text{0.817}}$ | $\underline{\text{0.685}}$ | 0.843 | 0.695 |
> > > > | Sdwpfm2 | 24 | **0.614** | **0.548** | $\underline{\text{0.683}}$ | $\underline{\text{0.603}}$ | 0.716 | 0.627 |
> > > > |  | 360 | **1.036** | **0.717** | 1.243 | 0.885 | $\underline{\text{1.125}}$ | $\underline{\text{0.837}}$ |
> > > > |  | Avg | **0.825** | **0.633** | 0.963 | 0.744 | $\underline{\text{0.921}}$ | $\underline{\text{0.732}}$ |
> > > > | Sdwpfh1 | 24 | **0.668** | **0.583** | $\underline{\text{0.696}}$ | $\underline{\text{0.641}}$ | 0.799 | 0.674 |
> > > > |  | 360 | **0.846** | **0.674** | 0.958 | 0.810 | $\underline{\text{0.937}}$ | $\underline{\text{0.779}}$ |
> > > > |  | Avg | **0.757** | **0.628** | $\underline{\text{0.827}}$ | $\underline{\text{0.725}}$ | 0.868 | 0.727 |
> > > > | Sdwpfh2 | 24 | $\underline{\text{0.797}}$ | **0.644** | **0.795** | $\underline{\text{0.687}}$ | 0.963 | 0.743 |
> > > > |  | 360 | **0.968** | **0.739** | $\underline{\text{1.025}}$ | $\underline{\text{0.790}}$ | 1.093 | 0.843 |
> > > > |  | Avg | **0.882** | **0.691** | $\underline{\text{0.910}}$ | $\underline{\text{0.738}}$ | 1.028 | 0.793 |
> > > > | Colbun | 10 | **0.064** | **0.081** | 0.115 | 0.156 | $\underline{\text{0.072}}$ | $\underline{\text{0.119}}$ |
> > > > |  | 30 | **0.168** | **0.246** | $\underline{\text{0.230}}$ | $\underline{\text{0.305}}$ | 0.329 | 0.380 |
> > > > |  | Avg | **0.116** | **0.164** | $\underline{\text{0.172}}$ | $\underline{\text{0.230}}$ | 0.201 | 0.249 |
> > > > | Rapel | 10 | **0.191** | **0.204** | $\underline{\text{0.224}}$ | $\underline{\text{0.233}}$ | 0.232 | 0.256 |
> > > > |  | 30 | $\underline{\text{0.325}}$ | **0.343** | 0.425 | 0.424 | **0.304** | $\underline{\text{0.352}}$ |
> > > > |  | Avg | **0.258** | **0.273** | 0.324 | 0.328 | $\underline{\text{0.268}}$ | $\underline{\text{0.304}}$ |

---

### Author Response · Authors · 2025-12-02
**Summary of Rebuttal (average rating 5.5 → 6.5)**

**Dear Reviewers, ACs, SACs, and PCs,**

We are sorry to hear about the recent OpenReview bug issue, and we fully support the proposed remedial actions.

At the same time, we emphasize that we have always followed the rules and have never exploited the OpenReview bug. Fortunately, thanks to the diligence and responsiveness of our reviewers, we had essentially concluded most meaningful discussions by **Nov. 26**, prior to the incident on **Nov. 27**.

To assist in the final assessment of our submission, we have summarized the consensus on our work's **strengths** and the **results of the discussion** below:

It is encouraging to see that reviewers agree on the following strengths of our work:

- **Important Problem & Practical Relevance**: The work addresses the critical challenge of time series forecasting with exogenous variables, which has practical applications across various domains. (soMm, wNXH)
- **Novel Unified Framework**: The first to jointly model time and channel dependencies, breaking the traditional two-step forecasting strategy and introducing a module to ensure predictions align with underlying temporal and channel correlations. (s4T3, wNXH, soMm)
- **High Effectiveness**: Extensive experiments demonstrate significant improvements over existing methods, with clear and well-presented figures. (vFgB, soMm, s4T3)
- **Clear Writing & Presentation**: The paper is well-structured, written clearly, and supported by informative figures that aid understanding. (vFgB, soMm)
- **Reproducibility & Openness**: Code and datasets are publicly available in an anonymous repository, supporting reproducibility and transparency. (wNXH, s4T3)

During the rebuttal phase, we managed to address the reviewers' concerns through:

- **Additional Experiments**: Conducted extensive experiments to further validate the soundness of our model design, the appropriateness of our baseline choices, and to ensure a more comprehensive evaluation.
- **Detailed Method Explanations**: Clarified the method presentation to help readers better understand our approach.
- **Manuscript Refinement**: Revised the manuscript to include updates and make it easier for readers to follow.

We are pleased that reviewers **soMm, s4T3 and vFgB** have explicitly confirmed that **their concerns were addressed**, supporting our work by **raising or maintaining their positive scores**. Reviewer **vFgB** not only **increased their score** but also provided additional suggestions, demonstrating a willingness to help us **further improve the quality of the paper**. We have **addressed these new suggestions with additional experiments**; unfortunately, the OpenReview incident prevented any further discussion.  Although reviewer **wNXH** did not respond, their initial score was positive. Therefore, after the rebuttal (before the OpenReview bug issue occurred), all reviewers expressed a positive view of our work.

In addition, we believe it is necessary to report the changes in our scores throughout the fruitful rebuttal phase up to **Nov. 27** (before the OpenReview bug issue broke out).

Table: Reviewers' Scores before/after Rebuttal up to **Nov. 27 (all changes can be traced in the corresponding comments)**.

| Reviewer | interaction | Rating | Confidence | Soundness | Presentation | Contribution |
| --- | --- | --- | --- | --- | --- | --- |
| vFgB | reviewers **increased** **rating** on **Nov. 23** and provided **further suggestion**; we **replied** **again**, but unfortunately could not further communicate | **2 -> 4**  | 4 -> 4 | **2 -> 3** | **2 -> 3** | **2 -> 3** |
| wNXH | no response so far | 6 -> 6 | 5 -> 5 | 3 -> 3 | 3 -> 3 | 4 -> 4 |
| soMm | reviewers **increased rating** on **Nov. 26** | **6 -> 8**  | 4 -> 4 | **3 -> 4** | 4 -> 4 | **3 -> 4** |
| s4T3 | reviewers replied and kept the score | 8 -> 8  | 5 -> 5 | 4 -> 4 | 4 -> 4 | 3 -> 3 |
| Avg. | - | **5.5 -> 6.5** | 4.5 -> 4.5 | **3 -> 3.5** | **3.25 -> 3.5** | **3 -> 3.5** |

Once again, we sincerely thank all reviewers for their efforts in reviewing our paper and for maintaining active communication with us throughout the rebuttal period.

Best regards,

Authors

---

### Meta-Review · Area_Chair_NEJo · 2025-12-24

**Summary:**

This paper introduces GCGNet, a generative approach for time-series forecasting that explicitly integrates exogenous variables and relational structures. The model first generates preliminary forecasts via a variational generator, then enforces correlation consistency through a graph-based discriminator, and finally refines the outputs with a dedicated graph refiner to mitigate degeneration. Experiments on 12 real-world datasets show that GCGNet consistently outperforms existing state-of-the-art forecasting methods.

**Reviewer Concerns:**

Most reviewers expressed concerns about the need for more extensive experimental validation to fully establish the soundness of GCGNet, as well as about the clarity and completeness of the method description and certain conceptual refinements. For instance, reviewer vFgB noted the absence of a thorough discussion of the challenges encountered in addressing the limitations of related work. Reviewer soMm pointed out that many of the selected baseline models do not naturally support exogenous variables, which may affect the fairness of the comparisons. Reviewer s4T3 noted that the description of the sparsification operation does not provide enough detail to clearly explain its underlying mechanism.

**Reviewer Scores:**

The initial overall scores from reviewers vFgB, wNXH, soMm, and s4T3 were 2, 6, 6, and 8, with corresponding confidence scores of 4, 5, 4, and 5. After reviewing the full rebuttal and the authors’ consolidated responses, most of the reviewers’ concerns were satisfactorily addressed, particularly those raised by reviewer vFgB regarding the validation of GCGNet’s soundness.

Based on the authors’ rebuttal and the reviewers’ participation in the discussion prior to the technical disruption, reviewers vFgB, soMm, and s4T3 provided positive assessments of the rebuttal. Overall, the authors’ responses effectively resolved the majority of concerns raised by the four reviewers.

---

### Decision · Program_Chairs · 2026-01-26

Accept (Poster)